# LEARNING-AUGMENTED SKETCHES FOR HESSIANS

## ABSTRACT

Sketching is a dimensionality reduction technique where one compresses a matrix by linear combinations that are chosen at random. A line of work has shown how to sketch the Hessian to speed up each iteration in a second order method, but such sketches usually depend only on the matrix at hand, and in a number of cases are even oblivious to the input matrix. One could instead hope to learn a distribution on sketching matrices that is optimized for the specific distribution of input matrices. We show how to design learned sketches for the Hessian in the context of second order methods. We prove that a smaller sketching dimension of the column space of a tall matrix is possible, given an oracle that can predict the indices of the rows of large leverage score. We design such an oracle for various datasets, and this leads to a faster convergence of the well-studied iterative Hessian sketch procedure, which applies to a wide range of problems in convex optimization. We show empirically that learned sketches, compared with their "non-learned" counterparts, do improve the approximation accuracy for important problems, including LASSO and matrix estimation with nuclear norm constraints.

## 1 INTRODUCTION

Large-scale optimization problems are abundant and solving them efficiently requires powerful tools to make the computation practical. This is especially true of second order methods which often are less practical than first order ones. Although second order methods may have many fewer iterations, each iteration could involve inverting a large Hessian, which is cubic time; in contrast, first order methods such as stochastic gradient descent are linear time per iteration.

In order to make second order methods faster in each iteration, a large body of work has looked at dimensionality reduction techniques, such as sampling, sketching, or approximating the Hessian by a low rank matrix. See, for example, (Gower et al., 2016; Xu et al., 2016; Pilanci & Wainwright, 2016; 2017; Doikov & Richtárik, 2018; Gower et al., 2018; Roosta-Khorasani & Mahoney, 2019; Gower et al., 2019; Kylasa et al., 2019; Xu et al., 2020; Li et al., 2020). Our focus is on sketching techniques, which often consist of multiplying the Hessian by a random matrix chosen independently of the Hessian. Sketching has a long history in theoretical computer science (see, e.g., (Woodruff, 2014) for a survey), and we describe such methods more below. A special case of sketching is sampling, which in practice is often uniform sampling, and hence oblivious to properties of the actual matrix. Other times the sampling is non-uniform, and based on squared norms of submatrices of the Hessian or on the leverage scores of the Hessian.

Our focus is on sketching techniques, and in particular, we consider the framework of (Pilanci & Wainwright, 2016; 2017) which introduces the iterative Hessian sketch and the Newton sketch, as well as the high accuracy refinement given in (van den Brand et al., 2020). If one were to run Newton's method to find a point where the gradient is zero, in each iteration one needs to solve an equation involving the current Hessian and gradient to find the update direction. When the Hessian can be decomposed as $A^\top A$ for an $n \times d$ matrix $A$ with $n \gg d$, then sketching is particularly suitable. The *iterative Hessian sketch* was proposed in Pilanci & Wainwright (2016), where $A$ is replaced with $S \cdot A$, for a random matrix $S$ which could be i.i.d. Gaussian or drawn from a more structured family of random matrices such as the Subsampled Randomized Hadamard Transforms or COUNT-SKETCH matrices; the latter was done in (Cormode & Dickens, 2019). The *Newton sketch* was proposed by Pilanci & Wainwright (2017), which extended sketching methods beyond constrained least-squares problems to any twice differentiable function subject to a closed convex constraint set. Using this sketch inside of interior point updates has led to much faster algorithms

for an extensive body of convex optimization problems (Pilanci & Wainwright, 2017). By instead using sketching as a preconditioner, an application of the work of (van den Brand et al., 2020) (see Appendix E) was able to improve the dependence on the accuracy parameter $\epsilon$ to logarithmic.

In general, the idea behind sketching is the following. One chooses a random matrix $S$, drawn from a certain family of random matrices, and computes $SA$. If $A$ is tall-and-thin, then $S$ is short-and-fat, and thus $SA$ is a small, roughly square matrix. Moreover, $SA$ preserves important properties of $A$. One typically desired property is that $S$ is a subspace embedding, meaning that $\|SAx\|_2 = (1 \pm \epsilon)\|Ax\|_2$ for all $x$ simultaneously. An observation exploited in Cormode & Dickens (2019), building off of the COUNT-SKETCH random matrices $S$ introduced in randomized linear algebra in Clarkson & Woodruff (2017), is that if $S$ contains a single non-zero entry per column, then $SA$ can be computed in $O(\mathrm{nnz}(A))$ time, where $\mathrm{nnz}(A)$ denotes the number of nonzeros in $A$. This is also referred to as input-sparsity running time.

Each iteration of a second order method often involves solving an equation of the form $A^\top A x = A^\top b$, where $A^\top A$ is the Hessian and $b$ is the gradient. For a number of problems, one has access to a matrix $A \in \mathbb{R}^{n \times d}$ with $n \gg d$, which is also an assumption made in Pilanci & Wainwright (2017). Therefore, the solution $x$ is the minimizer to a constrained least squares regression problem:

$$\min_{x \in \mathcal{C}} \frac{1}{2} \|Ax - b\|_2^2, \tag{1}$$

where $\mathcal{C}$ is a convex constraint set in $\mathbb{R}^d$. For the unconstrained case ($\mathcal{C} = \mathbb{R}^d$), various classical sketches that attain the subspace embedding property can provably yield high-accuracy approximate solutions (see, e.g., (Sarlos, 2006; Nelson & Nguyên, 2013; Cohen, 2016; Clarkson & Woodruff, 2017)); for the general constrained case, the Iterative Hessian Sketch (IHS) was proposed by Pilanci & Wainwright (2016) as an effective approach and Cormode & Dickens (2019) employed sparse sketches to achieve input-sparsity running time for IHS. All sketches used in these results are data-oblivious random sketches.

**Learned Sketching.** In the last few years, an exciting new notion of *learned sketching* has emerged. Here the idea is that one often sees independent samples of matrices $A$ from a distribution $\mathcal{D}$, and can train a model to learn the entries in a sketching matrix $S$ on these samples. When given a future sample $B$, also drawn from $\mathcal{D}$, the learned sketching matrix $S$ will be such that $S \cdot B$ is a much more accurate compression of $B$ than if $S$ had the same number of rows and were instead drawn without knowledge of $\mathcal{D}$. Moreover, the learned sketch $S$ is often *sparse*, therefore allowing $S \cdot B$ to be applied very quickly. For large datasets $B$ this is particularly important, and distinguishes this approach from other transfer learning approaches, e.g., (Andrychowicz et al., 2016), which can be considerably slower in this context.

Learned sketches were first used in the data stream context for finding frequent items (Hsu et al., 2019) and have subsequently been applied to a number of other problems on large data. For example, Indyk et al. (2019) showed that learned sketches yield significantly smaller errors for low rank approximation. Dong et al. (2020) made significant improvements to nearest neighbor search using learned sketches. More recently, Liu et al. (2020) extended learned sketches to several problems in numerical linear algebra, including least-squares regression, as well as $k$-means clustering.

Despite the number of problems that learned sketches have been applied to, they have not been applied to convex optimization in general. Given that such methods often require solving a large overdetermined least squares problem in each iteration, it is hopeful that one can improve each iteration using learned sketches. However, a number of natural questions arise: (1) *how should we learn the sketch?* (2) *should we apply the same learned sketch in each iteration, or learn it in the next iteration by training on a data set involving previously learned sketches from prior iterations?*

**Our Contributions.** In this work we answer the above questions and develop the first framework of learned sketching that applies to a wide number of problems in convex optimization. Namely, we apply learned sketches to constrained least-squares problems, including LASSO and matrix regression with nuclear norm constraints. We show empirically that learned sketches demonstrate superior accuracy over classical oblivious random sketches for each of these problems. All of our learned sketches $S$ are extremely sparse, meaning that they contain a single non-zero entry per column and that they can be applied in input-sparsity time. For such sketches, there are two things to learn: the position of the non-zero entry in each column and the value of the non-zero entry.

Following the previous work of Indyk et al. (2019), we choose the position of the nonzero entry in each column to be uniformly random, while the value of the nonzero entry is learned (the value is no longer limited to $-1$ and $1$). Here we consider a new learning objective, that is, we optimize the subspace embedding property of the sketching matrix instead of optimizing the error in the objective function of the optimization problem we are trying to solve. This demonstrates a significant advantage over non-learned sketches, and has a fast training time. Our experiments show that the convergence rate is reduced by $44\%$ over the nonlearned COUNT-SKETCH (a classical extremely sparse sketch) for the LASSO problem on a real-world dataset. Recall that a smaller convergence rate means a faster convergence.

We prove theoretically that $S$ can take fewer rows, with optimized positions of nonzero entries, when the input matrix $A$ has a small number of rows of heavy leverage score. More specifically, COUNT-SKETCH takes $O(d^2/(\delta\epsilon^2))$ rows with failure probability $\delta$, while our $S$ requires only $O((d\operatorname{polylog}(1/\epsilon) + \log(1/\delta))/\epsilon^2)$ rows if $A$ has at most $d\operatorname{polylog}(1/\epsilon)/\epsilon^2$ rows of leverage score at least $\epsilon/d$. This is a quadratic improvement in $d$ and an exponential improvement in $\delta$. Applying $S$ to $A$ runs in input-sparsity time and the resulting $SA$ may remain sparse if $A$ is sparse. In practice, it is not necessary to calculate the leverage scores. Instead, we show in our experiments that the indices of the rows of heavy leverage score can be learned and the induced $S$ achieves a comparable accuracy for the abovementioned LASSO problem to classical dense sketches such as Gaussian matrices.

Combining both aspects, the value of the nonzero entry and the indices of the rows of heavy leverage score, we obtain even better learned sketches. For the same LASSO problem, we show empirically that such learned sketches reduce the convergence rate by a larger $79.9\%$ to $84.6\%$ over non-learned sketches. Therefore, the learned sketches attain a smaller error within the same number of iterations, and in fact, within the same limit on the maximum runtime, since our sketches are extremely sparse.

We also study the general framework of convex optimization in van den Brand et al. (2020), and show that also for sketching-based preconditioning, learned sketches demonstrate considerable advantages. More precisely, by using a learned sketch with the same number of rows as an oblivious sketch, we are able to obtain a much better preconditioner with the same overall running time.

## 2 PRELIMINARIES

**Notation.** We denote by $\mathbb{S}^{n-1}$ the unit sphere in the $n$-dimensional Euclidean space $\mathbb{R}^n$. For a matrix $A \in \mathbb{R}^{m \times n}$ we denote by $\|A\|_{op}$ its operator norm, which is defined as $\|A\|_{\mathrm{op}} = \sup_{x \in \mathbb{S}^{n-1}} \|Ax\|_2$. We also denote by $\sigma_{\max}(A)$ and $\sigma_{\min}(A)$ the largest and smallest singular values of $A$, respectively, and by $\operatorname{colsp}(A)$ the column space of $A$. The condition number of $A$ is defined to be $\kappa(A) = \sigma_{\max}(A)/\sigma_{\min}(A)$.

---

**Algorithm 1** LEARN-SKETCH: Gradient descent algorithm for learning the sketch values

---

**Require:** $\mathcal{A}_{\text{train}} = \{A_i\}_{i=1}^N$ $(A_i \in \mathbb{R}^{n \times d})$, learning rate $\alpha$
1: Randomly initialize $p, v$ for a COUNT-SKETCH-type sketch as described in the text
2: **for** $t = 0$ **to** step **do**
3:     Form $S$ using $p, v$
4:     Sample batch $\mathcal{A}_{batch}$ from $\mathcal{A}_{train}$
5:     $v \leftarrow v - \alpha \frac{\partial \mathcal{L}(S, \mathcal{A}_{batch})}{\partial v}$

---

**Leverage Scores.** We only consider matrices of full column rank[1]. Suppose that $A \in \mathbb{R}^{m \times n}$ $(m \geq n)$ has full column rank. It has $m$ leverage scores, denoted by $\tau_1(A), \ldots, \tau_m(A)$, which are defined as $\tau_i(A) = \|e_i^\top A(A^\top A)^{-1} A^\top\|_2^2$, where $\{e_1, \ldots, e_m\}$ is the canonical basis of $\mathbb{R}^m$. Equivalently, letting $A = U\Sigma V^\top$ be the singular value decomposition of $A$, where $U \in \mathbb{R}^{m \times n}, \Sigma, V \in \mathbb{R}^{n \times n}$, we can also write $\tau_i(A) = \|e_i^\top UU^\top\|_2^2 = \|e_i^\top U\|_2^2$, which is the squared $\ell_2$ norm of the $i$-th row of $U$.

**Classical Sketches.** Below we review several classical sketches that have been used for solving optimization problems.

- Gaussian sketch: $S = \frac{1}{\sqrt{m}} G$, where $G \in \mathbb{R}^{m \times n}$ with i.i.d. $N(0,1)$ entries.

- COUNT-SKETCH: Each column of $S$ has only a single non-zero entry. The position of the non-zero entry is chosen uniformly over the $m$ entries in the column and the value of the entry is either $+1$ or $-1$, each with probability $1/2$. Further, the columns are chosen independently.

- Sparse Johnson-Lindenstrauss Transform (SJLT): $S$ is the vertical concatenation of $s$ independent COUNT-SKETCH matrices, each of dimension $m/s \times n$.

---

[1]This can be assumed w.l.o.g. by adding artbirarily small random noise to the input, or one can first quickly use sketching to find a subset of columns of maximum rank, and replace the inut with that subset of columns.

**COUNT-SKETCH-type Sketch.** A COUNT-SKETCH-type sketch is characterized by a tuple $(m, n, p, v)$, where $m, n$ are positive integers and $p, v$ are $n$-dimensional real vectors, defined as follows. The sketching matrix $S$ has dimensions $m \times n$ and $S_{p_i,i} = v_i$ for all $1 \leq i \leq n$, while all the other entries of $S$ are 0. When $m$ and $n$ are clear from context, we may characterize such a sketching matrix by $(p, v)$ only.

**Subspace Embeddings.** For a matrix $A \in \mathbb{R}^{n \times d}$, we say a matrix $S \in \mathbb{R}^{m \times n}$ is a $(1 \pm \epsilon)$-subspace embedding for the column span of $A$ if $(1 - \epsilon) \|Ax\|_2 \leq \|SAx\|_2 \leq (1 + \epsilon) \|Ax\|_2$ for all $x \in \mathbb{R}^d$. The classical sketches above, with appropriate parameters, are all subspace embedding matrices with probability at least $1 - \delta$; our focus is on COUNT-SKETCH which can be applied in input sparsity running time. We summarize the parameters needed for a subspace embedding below:

- Gaussian sketch: $m = O((d + \log(1/\delta))/\epsilon^2)$. It is a dense matrix and computing $SA$ costs $O(m \cdot \mathrm{nnz}(A)) = O(\mathrm{nnz}(A)(d + \log(1/\delta))/\epsilon^2)$ time.
- COUNT-SKETCH: $m = O(d^2/(\delta\epsilon^2))$ (Clarkson & Woodruff, 2017). Though the number of rows is quadratic in $d/\epsilon$, the matrix $S$ is sparse and computing $SA$ takes only $O(\mathrm{nnz}(A))$ time.
- SJLT: $m = O(d \log(\frac{d}{\delta})/\epsilon^2)$ and has $s = O(\log(\frac{d}{\delta})/\epsilon)$ non-zeros per column (Nelson & Nguyên, 2013; Cohen, 2016). Computing $SA$ takes $O(s \, \mathrm{nnz}(A)) = O(\mathrm{nnz}(A) \log(\frac{d}{\delta})/\epsilon)$ time.

**Iterative Hessian Sketch.** The Iterative Hessian Sketching (IHS) method (Pilanci & Wainwright, 2016) solves the constrained least-squares problem (1) by iteratively performing the update

$$x_{t+1} = \arg\min_{x \in \mathcal{C}} \left\{ \frac{1}{2} \|S_{t+1}A(x - x_t)\|_2^2 - \langle A^\top(b - Ax_t), x - x_t \rangle \right\}, \tag{2}$$

where $S_{t+1}$ is a sketching matrix. It is not difficult to see that for the unsketched version ($S_{t+1}$ is the identity matrix) of the minimization above, the optimal solution $x^{t+1}$ coincides with the optimal solution to the constrained least squares problem (1). The IHS approximates the Hessian $A^\top A$ by a sketched version $(S_{t+1}A)^\top(S_{t+1}A)$ to improve runtime, as $S_{t+1}A$ typically has very few rows.

**Unconstrained Convex Optimization.** Consider an unconstrained convex optimization problem $\min_x f(x)$, where $f$ is smooth and strongly convex, and its Hessian $\nabla^2 f$ is Lipschitz continuous. This problem can be solved by Newton's method, which iteratively performs the update

$$x_{t+1} = x_t - \arg\min_z \left\| (\nabla^2 f(x_t)^{1/2})^\top (\nabla^2 f(x_t)^{1/2})z - \nabla f(x_t) \right\|_2, \tag{3}$$

provided it is given a good initial point $x_0$. In each step, it requires solving a regression problem of the form $\min_z \|A^\top A z - y\|_2$, which, with access to $A$, can be solved with a fast regression solver in (van den Brand et al., 2020). The regression solver first computes a preconditioner $R$ via a QR decomposition such that $SAR$ has orthonormal columns, where $S$ is a sketching matrix, then solves $\widehat{z} = \arg\min_{z'} \|(AR)^\top(AR)z' - y\|_2$ by gradient descent and returns $R\widehat{z}$ in the end. Here, the point of sketching is that the QR decomposition of $SA$ can be computed much more efficiently than the QR decomposition of $A$, since $S$ has only a small number of rows.

**Learning a Sketch.** We use the same learning algorithm in (Liu et al., 2020), given in Algorithm 1. The algorithm aims to minimize the mean loss function $\mathcal{L}(S, \mathcal{A}) = \frac{1}{N} \sum_{i=1}^{N} \mathcal{L}(S, A_i)$, where $S$ is the learned sketch, $\mathcal{L}(S, A)$ is the loss function of $S$ applied to a data matrix $A$, and $\mathcal{A} = \{A_1, \dots, A_N\}$ is a (random) subset of training data.

## 3 LEARNING-AUGMENTED SUBSPACE EMBEDDINGS

In this section we explain two ways to optimize the subspace embedding property of the sketching matrix. One is to optimize the non-zero positions of the COUNT-SKETCH-type sketch, based on a trained oracle to identify a superset of the rows of large leverage score. The other is to optimize the values of the nonzero entries, which may no longer be $-1$ or $1$, via a learning algorithm based on gradient descent. As we shall see in Section 4 and 5, a better subspace embedding implies a better convergence rate in the IHS, as well as for the subroutine in unconstrained convex optimization.

### 3.1 SKETCHED LEARNING: OPTIMIZING THE POSITIONS

In this section we consider the problem of embedding the column space of a matrix $A \in \mathbb{R}^{n \times d}$, provided that $A$ has a few rows of large leverage score, as well as access to an oracle which reveals a

*superset* of the indices of such rows. Formally, let $\tau_i(A)$ denote the leverage score of the $i$-th row of $A$ and let

$$I^* = \{i : \tau_i(A) \geq \nu\}$$

be the set of rows with large leverage score. Suppose that a superset $I \supseteq I^*$ is known to the algorithm. In the experiments we train an oracle to predict such rows. We can maintain all rows in $I$ explicitly and apply a COUNT-SKETCH to the remaining rows, i.e., the rows in $[n] \setminus I$. Up to permutation of the rows, we can write

$$A = \begin{pmatrix} A_I \\ A_{I^c} \end{pmatrix} \quad \text{and} \quad S = \begin{pmatrix} I & 0 \\ 0 & S' \end{pmatrix}, \tag{4}$$

where $S'$ is a random COUNT-SKETCH matrix of $m$ rows. Clearly $S$ has a single non-zero entry per column. We have the following theorem, whose proof is postponed to Section A. Intuitively, the proof for COUNT-SKETCH in (Clarkson & Woodruff, 2017) handles rows of large leverage score and rows of small leverage score separately. The rows of large leverage score are to be perfectly hashed while the rows of small leverage score will concentrate in the sketch by the Hanson-Wright inequality.

**Theorem 3.1.** *Let $\nu = \epsilon/d$. Suppose that $m = O((d/\epsilon^2)(\mathrm{polylog}(1/\epsilon) + \log(1/\delta)))$, $\delta \in (0, 1/m]$ and $d = \Omega((1/\epsilon)\,\mathrm{polylog}(1/\epsilon)\log^2(1/\delta))$. Then, there exists a distribution on $S$ of the form in (4) with $m + |I|$ rows such that $\Pr\left\{\forall x \in \mathrm{colsp}(A), \left| \|Sx\|_2^2 - \|x\|_2^2 \right| > \epsilon \|x\|_2^2 \right\} \leq \delta$.*

Hence, if there happen to be at most $d\,\mathrm{polylog}(1/\epsilon)/\epsilon^2$ rows of leverage score at least $\epsilon/d$, the overall sketch length for embedding $\mathrm{colsp}(A)$ can be reduced to $O((d\,\mathrm{polylog}(1/\epsilon) + \log(1/\delta))/\epsilon^2)$, a quadratic improvement in $d$ and an exponential improvement in $\delta$ over the original sketch length of $O(d^2/(\epsilon^2\delta))$ for COUNT-SKETCH. In the worst case there could be $O(d^2/\epsilon)$ such rows, though empirically we do not observe this. The following is an immediate corollary, by setting $\delta = 1/m$.

**Corollary 3.2.** *Suppose that $d = \Omega((1/\epsilon)\,\mathrm{polylog}(1/\epsilon))$ and $|I| = O((d/\epsilon^2)\,\mathrm{polylog}(d/\epsilon))$ with $\nu = \epsilon/d$. There exists a distribution on $S$ of the form in (4) with $O((d/\epsilon^2)\,\mathrm{polylog}(d/\epsilon))$ rows such that $\Pr\left\{\forall x \in \mathrm{colsp}(A), \left| \|Sx\|_2^2 - \|x\|_2^2 \right| > \epsilon \|x\|_2^2 \right\} \leq \epsilon^3$.*

We remark that our $S$ is of the COUNT-SKETCH type, which has a twofold benefit. First, $SA$ can be applied in $O(\mathrm{nnz}(A))$ time. This is faster than a chained subspace embedding of the form $S_2 S_1 A$, where $S_1$ is a COUNT-SKETCH matrix of $O(d^2/\epsilon^2)$ rows and $S_2$ is a subspace embedding matrix of $O(d/\epsilon^2)$ rows. Computing $S_1 A$ takes $O(\mathrm{nnz}(A))$ time but computing $S_2(S_1 A)$ will take an additional time of $\mathrm{poly}(d/\epsilon)$ or $O(\mathrm{nnz}(S_1 A)\log(d)/\epsilon)$. The latter terms can be quite large and even comparable to $n$ if say, $n$ is close to $d^2$. Second, our $S$ allows the sketched matrix $SA$ to be sparse when $A$ is sparse, while the other designs such as Subsampled Randomized Hadamard Transforms and Sparse Johnson-Lindentrauss Transforms either would not guarantee that $SA$ is sparse, or would yield a worse sparsity than a matrix of the COUNT-SKETCH type. The sparsity of $SA$ is also important for solving regression problems involving $B = SA$ in intermediate steps, as algorithms such as conjugate gradient, which use matrix-vector products, become more efficient.

We note that approximate leverages scores of all rows can be found in time $O(\mathrm{nnz}(A)\log n + \mathrm{poly}(d/\epsilon))$ (Clarkson & Woodruff, 2017). Hence, one can approximate the leverage score of every row in a preprocessing step before running the IHS. This time will be amortized by the IHS iterations, because the matrix $A$ remains the same throughout the process. Moreover, in Section 6, we show that for a number of real-world datasets, it is possible to learn the indices of the heavy rows. In practice, one can shrink the size of the superset $I$ by restricting $I$ to the rows with large $\ell_2$ norms in $A_I$. We shall demonstrate in Section 6 that this heuristic works well on some real-world datasets.

## 3.2 SKETCHED LEARNING: OPTIMIZING THE VALUES

As mentioned in Section 2, when we fix the positions of the non-zero entries, we aim to optimize the values by gradient descent. We propose the following objective loss function for the learning algorithm $\mathcal{L}(S, A_i) = \|(A_i R_i)^\top A_i R_i - I\|_F$, over all the training data, where $R_i$ comes from the QR-decomposition of $SA_i = Q_i R_i^{-1}$. We found empirically that not squaring this loss function works better than squaring it. We think one of the reasons is that the version without squaring may be less sensitive to outliers. The intuition for this loss function is given by the lemma below, whose proof is deferred to Section B.

**Lemma 3.3.** *Suppose that $\epsilon \in (0, \frac{1}{2})$, $S \in \mathbb{R}^{m \times n}$, $A \in \mathbb{R}^{n \times d}$ has full column rank, and $SA = QR$ is the QR-decomposition of $SA$. If $\|(AR^{-1})^\top AR^{-1} - I\|_{\mathrm{op}} \leq \epsilon$, then $S$ is a $(1 \pm \epsilon)$-subspace embedding of the column space of $A$.*

Lemma 3.3 implies that if the loss function over $\mathcal{A}_{\mathrm{train}}$ is small and the distribution of $\mathcal{A}_{\mathrm{test}}$ is similar to $\mathcal{A}_{\mathrm{train}}$, it is reasonable to expect that $S$ is a good subspace embedding of $\mathcal{A}_{\mathrm{test}}$. Here we use the Frobenius norm rather than operator norm in the loss function because it will make the optimization problem easier to solve, and our empirical results also show that the performance of the Frobenius norm is better than that of the operator norm.

## 4 HESSIAN SKETCH

In this section, we consider the minimization problem

$$\min_{x \in \mathcal{C}} \left\{ \frac{1}{2} \|SAx\|_2^2 - \langle A^\top y, x \rangle \right\}, \qquad (5)$$

which is used as a subroutine for the IHS (cf. (2)). We present an algorithm with the learned sketch in Algorithm 2. To analyze its performance, we define the following quantities (corresponding exactly to the unconstrained case in (Pilanci & Wainwright, 2016))

---

**Algorithm 2** Solver for (5)

1: $S_1 \leftarrow$ learned sketch, $S_2 \leftarrow$ random sketch
2: $(\widehat{Z}_{i,1}, \widehat{Z}_{i,2}) \leftarrow \text{ESTIMATE}(S_i, A)$, $i = 1, 2$
3: $i^* \leftarrow \arg\min_{i=1,2}(\widehat{Z}_{i,2}/\widehat{Z}_{i,1})$
4: $\widehat{x} \leftarrow$ solution of (5) with $S = S_{i^*}$
5: **return** $\widehat{x}$

6: **function** ESTIMATE($S, A$)
7:     $T \leftarrow$ sparse $(1\pm\eta)$-subspace embedding matrix for $d$-dimensional subspaces
8:     $(Q, R) \leftarrow \text{QR}(TA)$
9:     $\widehat{Z}_1 \leftarrow \sigma_{\min}(SAR^{-1})$
10:     $\widehat{Z}_2 \leftarrow (1 \pm \eta)$-approximation to $\left\| (SAR^{-1})^\top (SAR^{-1}) - I \right\|_{\mathrm{op}}$
11:     **return** $(\widehat{Z}_1, \widehat{Z}_2)$

---

$$Z_1(S) = \inf_{v \in \mathrm{colsp}(A) \cap \mathbb{S}^{n-1}} \|Sv\|_2^2, \quad Z_2(S) = \sup_{u,v \in \mathrm{colsp}(A) \cap \mathbb{S}^{n-1}} \langle u, (S^\top S - I_n)v \rangle.$$

When $S$ is a $(1 + \epsilon)$-subspace embedding of $\mathrm{colsp}(A)$, we have $Z_1(S) \geq 1 - \epsilon$ and $Z_2(S) \leq 2\epsilon$.

For a general sketching matrix $S$, the following is the approximation guarantee of $\widehat{Z}_1$ and $\widehat{Z}_2$, which are estimates of $Z_1(S)$ and $Z_2(S)$, respectively. The proof is postponed to Appendix C. The main idea is that $AR^{-1}$ is well-conditioned, where $R$ is as calculated in Algorithm 2.

**Lemma 4.1.** *Suppose that $\eta \in (0, \frac{1}{3})$ is a small constant, $A$ is of full rank and $S$ has $\mathrm{poly}(d/\eta)$ rows. The function ESTIMATE($S, A$) returns in $O((\mathrm{nnz}(A) \log \frac{1}{\eta} + \mathrm{poly}(\frac{d}{\eta}))$ time $\widehat{Z}_1, \widehat{Z}_2$ which with probability at least $0.99$ satisfy that $\frac{Z_1(S)}{1+\eta} \leq \widehat{Z}_1 \leq \frac{Z_1(S)}{1-\eta}$ and $\frac{Z_2(S)}{(1+\eta)^2} - 3\eta \leq \widehat{Z}_2 \leq \frac{Z_2(S)}{(1-\eta)^2} + 3\eta$.*

Similar to Proposition 1 of (Pilanci & Wainwright, 2016), we have the following guarantee. The proof is postponed to Appendix D.

**Theorem 4.2.** *Let $\eta \in (0, \frac{1}{3})$ be a small constant. Suppose that $A$ is of full rank and $S_1$ and $S_2$ are both COUNT-SKETCH-type sketches with $\mathrm{poly}(d/\eta)$ rows. Algorithm 2 returns a solution $\widehat{x}$ which, with probability at least $0.98$, satisfies that $\|A(\widehat{x} - x^*)\|_2 \leq (1 + \eta)^4 \left( \min \left\{ \frac{\widehat{Z}_{1,2}}{\widehat{Z}_{1,1}}, \frac{\widehat{Z}_{2,2}}{\widehat{Z}_{2,1}} \right\} + 4\eta \right) \|Ax^*\|_2$ in $O(\mathrm{nnz}(A) \log(\frac{1}{\eta}) + \mathrm{poly}(\frac{d}{\eta}))$ time, where $x^* = \arg\min_{x \in \mathcal{C}} \|Ax - b\|_2$ is the least-squares solution.*

Theorem 4.2 suggests the following. If the ratio of the learned sketch $Z_2(S_1)/Z_1(S_1)$ is a constant smaller than that of the random sketch $Z_2(S_2)/Z_2(S_2)$ and $\eta$ is a constant fraction of the ratio gap, then $\widehat{Z}_{1,2}/\widehat{Z}_{1,1}$ is a constant smaller than $\widehat{Z}_{2,2}/\widehat{Z}_{2,1}$, which means that the procedure of IHS will converge faster with the learned sketch. In particular, if $S_i$ is a $(1 + \epsilon_i)$-subspace embedding matrix for $\mathrm{colsp}(A)$ with $\epsilon_i < 1/3$ and $\eta < \gamma \min\{|\epsilon_1 - \epsilon_2|, \epsilon_1, \epsilon_2\}$ for some small constant $\gamma > 0$, we have $Z_2(S_i)/Z_1(S_i) \leq 3\epsilon_i$ and the guarantee in Theorem 4.2 becomes $\|A(\widehat{x} - x^*)\|_2 \leq O(\min\{\epsilon_1, \epsilon_2\}) \|Ax^*\|_2$, that is, a better subspace embedding can lead to a faster convergence. Hence, if the learned sketch is a better subspace embedding than a random sketch, theoretically we can obtain a better convergence by setting $\eta$ small enough; in practice we shall observe this.

Furthermore, if we know the indices of the rows of large leverage scores of $A$ and the assumptions in Corollary 3.2 are satisfied, we can use $O(d^2/\epsilon^2)$ rows to obtain a $\left(1 + O(\frac{\epsilon}{\sqrt{d}/\mathrm{polylog}(d/\epsilon)})\right)$-subspace embedding using Corollary 3.2, which is almost a $\sqrt{d}$-factor better than the usual guarantee

of a random COUNT-SKETCH matrix of the same dimension, leading to an algorithm of faster convergence.

## 5 HESSIAN REGRESSION

In this section, we consider the minimization problem

$$\min_z \left\| A^\top A z - y \right\|_2, \qquad (6)$$

which is used as a subroutine for the unconstrained convex optimization problem $\min_x f(x)$ with $A^\top A$ being the Hessian matrix $\nabla^2 f(x)$ (see (3)). Here $A \in \mathbb{R}^{n \times d}$, $y \in \mathbb{R}^d$, and we have access to $A$. We incorporate a learned sketch into the fast regression solver in (van den Brand et al., 2020) and present the algorithm in Algorithm 3.

---

**Algorithm 3** Fast Regression Solver for (6)

1: $S_1 \leftarrow$ learned sketch, $S_2 \leftarrow$ random sketch
2: $(Q_i, R_i) \leftarrow \text{QR}(S_i A)$, $i = 1, 2$
3: $(\sigma_i, \sigma_i') \leftarrow \text{EIG}(A R_i^{-1})$, $i = 1, 2 \, \triangleright \text{EIG}(B)$ returns estimates of $\sigma_{\max}(B)$ and $\sigma_{\min}(B)$
4: $i^* \leftarrow \min_{i=1,2}(\sigma_i / \sigma_i')$
5: $P \leftarrow R_{i^*}^{-1}$
6: $\eta \leftarrow 1/(\sigma_{i^*}^2 + (\sigma_{i^*}')^2)$
7: $z_0 \leftarrow 0$
8: **while** $\left\| A^\top A P z_t - y \right\|_2 \geq \epsilon \left\| y \right\|_2$ **do**
9: $\quad z_{t+1} \leftarrow z_t - \eta(P^\top A^\top A P)(P^\top A^\top A P z_t - P^\top y)$
10: **return** $P z_t$

---

Here the subroutine $\text{EIG}(B)$ applies a $(1 + \eta)$-subspace embedding sketch $T$ to $B$ for some small constant $\eta$ and returns $\sigma_{\max}(TB)$ and $\sigma_{\min}(TB)$. Since $B$ admits the form of $AR$, the sketched matrix $TB$ can be calculated as $(TA)R$ and thus can be computed in $O(\text{nnz}(A) + \text{poly}(d))$ time if $T$ is a COUNT-SKETCH matrix of $O(d^2)$ rows. The extreme singular values of $TB$ can be found by SVD or the Lanczos algorithm.

Similar to Lemma 4.2 in (van den Brand et al., 2020), we have the following guarantee of Algorithm 3. The proof parallels the proof in (van den Brand et al., 2020) and is postponed to Appendix E.

**Theorem 5.1.** *Suppose that $S_1$ and $S_2$ are both* COUNT-SKETCH-*type sketches with $O(d^2)$ rows. Algorithm 3 returns a solution $x'$ such that $\|A^\top A x' - y\|_2 \leq \epsilon \|y\|_2$ with probability at least $0.97$. The runtime is $O(\text{nnz}(A)) + \widetilde{O}(nd \cdot (\min\{\sigma_1/\sigma_1', \sigma_2/\sigma_2'\})^2 \cdot \log(\kappa(A)/\epsilon) + \text{poly}(d))$.*

**Remark 5.2.** In Algorithm 3, $S_2$ can be chosen to be a subspace embedding matrix for $d$-dimensional subspaces, in which case, $AR_2^{-1}$ has condition number close to 1 (see, e.g., p38 of (Woodruff, 2014)) and the full algorithm would run faster than the trivial $O(nd^2)$-time solver to (6).

**Remark 5.3.** For the original unconstrained convex optimization problem $\min_x f(x)$, one can run the entire optimization procedure with learned sketches versus the entire optimization procedure with random sketches, compare the objective values at the end, and choose the better of the two. For least-squares, $f(x) = \frac{1}{2} \left\| Ax - b \right\|_2^2$, and the value of $f(x)$ can be approximated efficiently by a sparse subspace embedding matrix in $O(\text{nnz}(A) + \text{nnz}(b) + \text{poly}(d))$ time.

## 6 EXPERIMENTS

**Comparison.** We compare the learned sketch against three classical sketches: Gaussian, COUNT-SKETCH, and SJLT (see Section 2) in all experiments. The quantity we compare is a certain error, defined individually for each problem, in each iteration of the IHS or the internal regression problem in fast regression. All of our experiments are conducted on a laptop with a 1.90GHz CPU and 16GB RAM. The offline training is done separately and the training of a single sketch matrix in our dataset can be finished within 5 minutes using a single GPU. For the learned sketches with learned values of nonzero entries, we take an average over three independent trials; for all other sketches, we take an average over five independent trials. The details of the implementation are deferred to Appendix H.

We elaborate on the reason that the horizontal axes in the plots are in terms of iterations rather than in terms of runtime. The learned matrix $S$ is trained offline only once using the training data. It is not computed while solving the optimization problem on the test data. *Hence, no additional computational cost is incurred in generating $S$ other than solving the iteration step using* COUNT-SKETCH. Since Gaussian matrices and sparse JL transforms are denser than COUNT-SKETCH matrices, they will be considerably slower in each round. Since we want to understand the convergence behavior, an iteration count is more revealing than an overall time bound. If our learned sketch performs no worse with respect to the total number of rounds (which our experiments show), then it has an even greater advantage in runtime. To substantiate this claim, we show in Appendix F an error-versus-runtime plot for the task of matrix estimation with nuclear norm constraints.

### 6.1 IHS EXPERIMENTS: LASSO

We define an instance of LASSO regression to be:

$$x^* = \arg\min_{\|x\|_1 \leq \lambda} \frac{1}{2} \|Ax - b\|_2^2, \tag{7}$$

where $\lambda$ is a parameter. We use two real-world datasets:

- **Electric**[2]: residential electric load measurements. Each row of the matrix corresponds to a different residence. Matrix columns are consecutive measurements from different times. $A_i \in \mathbb{R}^{370 \times 9}$, $b_i \in \mathbb{R}^{370 \times 1}$, and $|(A, b)_{\text{train}}| = 320$, $|(A, b)_{\text{test}}| = 80$. We set $\lambda = 15$.
- **Greenhouse gas** (GHG)[3]: time series of measured greenhouse gas concentrations in the California atmosphere. Each $(A, b)$ corresponds to a different measurement location. $A_i \in \mathbb{R}^{327 \times 14}$, $b_i \in \mathbb{R}^{327 \times 1}$, and $|(A, b)_{\text{train}}| = 400$, $|(A, b)_{\text{test}}| = 100$. We set $\lambda = 30$.

**Experiment Setting**: We choose $m = 6d, 8d, 10d$ for both datasets. We consider the error $\frac{1}{2} \|Ax - b\|_2^2 - \frac{1}{2} \|Ax^* - b\|_2^2$. For the two datasets, we use both the methods proposed in Section 3. For the heavy-row Count-Sketch, we allocate 30% of the sketch space to the rows of heavy leverage score. For the Electric dataset, each row represents a specific residence and the indices of the heavy rows do not vary much across the matrices in the training data. We select the heavy rows according to the number of times each row is heavy in the training data for the heavy rows. We also consider optimizing the non-zero values after identifying the heavy rows. For the GHG dataset, each row represents a specific time point and the heavy rows are not very concentrated. Nevertheless, we can find a superset of about 30% of the rows that contains most of the heavy rows, based on the counts on the training data. Then we prune the superset by selecting the rows with the largest $\ell_2$ norms, subject to the dimension budget. This will incur an additional computational cost, but the time is almost the same as the time to read the sub-matrix of these rows, and it can be used in all iterations, so the time of this step is negligible compared to the total runtime. We might lose a small fraction of heavy rows, but it only negligibly affects the experiments. The distribution on the indices of the heavy rows over the dataset is discussed in Appendix G.

**Experimental Result**: We plot in a logarithmic scale the mean errors of the two datasets in Figures 1 and 2. We see all methods display linear convergence, that is, letting $e_k$ denote the error in the $k$-th iteration, we have $e_k \approx \rho^k e_1$ for some convergence rate $\rho$. A smaller convergence rate implies a faster convergence.

We calculate an estimated rate of convergence $\rho = (e_k/e_1)^{1/k}$ with $k = 10$ for the GHG dataset, and with $k = 7$ for the Electric dataset. For the GHG dataset, we can see that when the sketch size is small ($m = 6d$), the gradient-based learned sketch has a rate of convergence that is 56% of that of COUNT-SKETCH, and the heavy-rows sketch has a convergence rate that is 86.9%. When the sketch size is large ($m = 10d$), the gradient-based learned sketch has a convergence rate that is 63.7%, and the heavy-rows sketch is 82.1%. For the Electric dataset, both sketches, especially the heavy-rows sketch, show significant improvements. When the sketch size is small, the combined-learned sketch has a convergence rate that is just 21.1% of that of sparse JL, and when the sketch size is large, the combined-learned sketch has a smaller convergence rate that is just 15.4%.

We also conducted IHS experiments for the matrix estimation problem with a nuclear norm constraint in Appendix F.

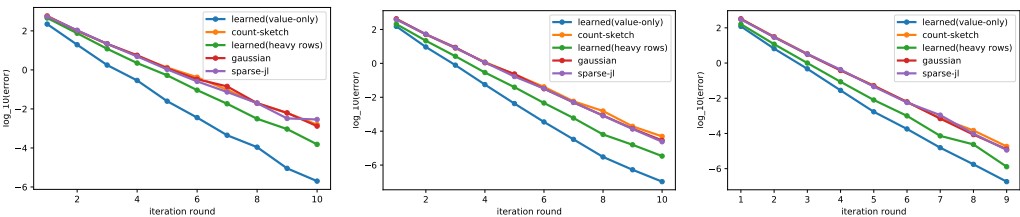

Figure 1: Test error of LASSO in the Green House Gas dataset.

---

[2]https://archive.ics.uci.edu/ml/datasets/ElectricityLoadDiagrams20112014
[3]https://archive.ics.uci.edu/ml/datasets/Greenhouse+Gas+Observing+Network

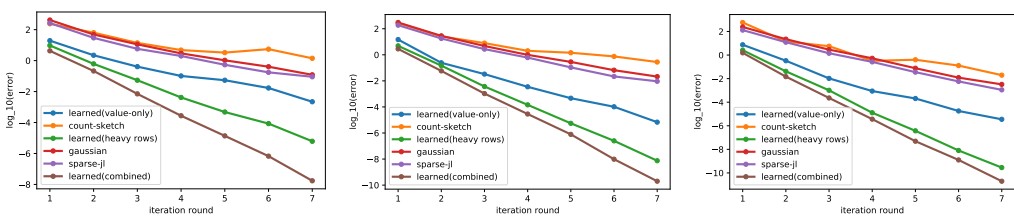

Figure 2: Test error of LASSO in Electric dataset.

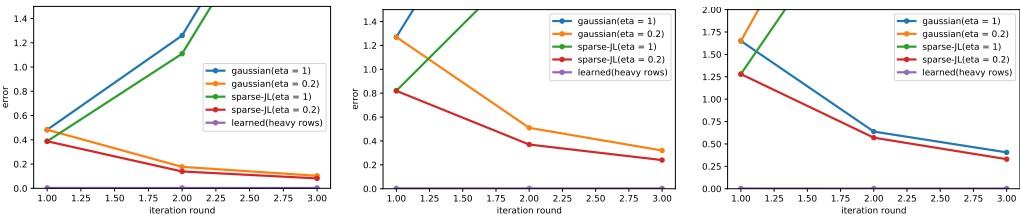

Figure 3: Test error of the subroutine in fast regression on Electric dataset.

## 6.2 FAST REGRESSION EXPERIMENTS

We consider the unconstrained least squares problem $\min_x f(x)$ with $f(x) = \frac{1}{2} \|Ax - b\|_2^2$ using the Electric dataset.

**Training**: Note that $\nabla^2 f(x) = A^\top A$, independent of $x$. In the $t$-th round of Newton's method, by (3), we need to solve a regression problem $\min_z \|A^\top A z - y\|_2^2$ with $y = \nabla f(x_t)$. Hence, we can use the same two methods in the preceding subsection to optimize the learned sketch $S_i$. For a general problem where $\nabla^2 f(x)$ depends on $x$, one can take $x_t$ to be the solution obtained from Algorithm 3 using the learned sketch $S_t$ to generate $A$ and $y$ for the $(t+1)$-st round, train a learned sketch $S_{t+1}$, and repeat this process.

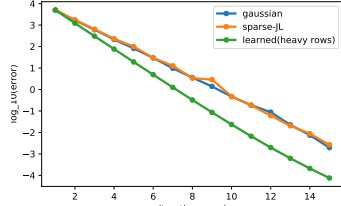

Figure 4: Test error of fast regression on Electric dataset

**Setup for Experiments**: For the Electric dataset, we set $m = 10d = 90$. We compare the heavy-rows COUNT-SKETCH matrix with the three classical random sketches, COUNT-SKETCH, Gaussian and Sparse-JL. For the parameter $\eta$ in Algorithm 3, we set $\eta = 1$ in all iterations for heavy-rows sketches. For the classical random sketches, we set $\eta$ in the following two ways: (a) $\eta = 1$ in all iterations and (b) $\eta = 1$ in the first iteration and $\eta = 0.2$ in all subsequent iterations.

**Experimental Results**: We examine the accuracy of the subproblem (6) and define the error to be $\|A^\top A R z_t - y\|_2 / \|y\|_2$. We consider the subproblems in the first three iterations of the global Newton method. The results are plotted in Figure 3. In this task, the COUNT-SKETCH causes a terrible divergence of the subroutine and is thus omitted in the plots. Still, we observe that in setting (a) of $\eta$, the other two classical sketches cause the subroutine to diverge. In setting (b) of $\eta$, the other two classical sketches lead to convergence but their error is significantly larger than that of the heavy-rows sketches, in each of the first three calls to the subroutine. The error of the heavy-rows sketch is less than $0.01$ in all iterations of all three subroutine calls, in both setting (a) and (b) of $\eta$.

We also plot a figure on the convergence of the global Newton method. Here, for each subroutine, we only run one iteration, and plot the error of the original least squares problem. The result is shown in Figure 4, which clearly displays a significantly faster decay with heavy-rows sketches. The rate of convergence using heavy-rows sketches is $80.6\%$ of that using Gaussian or sparse JL sketches.

**CONCLUSION.** We demonstrated the superiority of using learned sketches over classical random sketches, for the Iterative Hessian Sketch method which is used for a number of problems in convex optimization. Compared with random sketches, our learned sketches of the same size yield considerably faster convergence. We also provably show a better subspace embedding property of a sketch of the same size given an oracle for predicting a superset of rows with large leverage score. Our experiments show the construction of such an oracle is possible for real data sets, and they demonstrate a significant advantage over non-learned sketches for problems in convex optimization.

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

## A    PROOF OF THEOREM 3.1

First we prove the following lemma.

**Lemma A.1.** *Let $\delta \in (0, 1/m]$. It holds with probability at least $1 - \delta$ that*

$$\sup_{x \in \mathrm{colsp}(A)} \left| \|Sx\|_2^2 - \|x\|_2^2 \right| \leq \epsilon \|x\|_2^2,$$

*provided that*

$$m \gtrsim \epsilon^{-2}((d + \log m)\min\{\log^2(d/\epsilon), \log^2 m\} + d\log(1/\delta)),$$
$$1 \gtrsim \epsilon^{-2}\nu((\log m)\min\{\log^2(d/\epsilon), \log^2 m\} + \log(1/\delta))\log(1/\delta).$$

*Proof.* We shall adapt the proof of Theorem 5 in (Bourgain et al., 2015) to our setting. Let $T$ denote the unit sphere in $\mathrm{colsp}(A)$ and set the sparsity parameter $s = 1$. Observe that $\|Sx\|_2^2 = \|x_I\|_2^2 + \|Sx_{I^c}\|_2^2$, and so it suffices to show that

$$\Pr\left\{ \left| \|S'x_{I^c}\|_2^2 - \|x_{I^c}\|_2^2 \right| > \epsilon \right\} \leq \delta$$

for $x \in T$. We make the following definition, as in (2.6) of Bourgain et al. (2015):

$$A_{\delta,x} := \sum_{i=1}^m \sum_{j \in I^c} \delta_{ij} x_j e_i \otimes e_j,$$

and thus, $S'x_{I^c} = A_{\delta,x}\sigma$. Also by $\mathbb{E}\|S'x_{I^c}\|_2^2 = \|x_{I^c}\|_2^2$, one has

$$\sup_{x \in T} \left| \|S'x_{I^c}\|_2^2 - \|x_{I^c}\|_2^2 \right| = \sup_{x \in T} \left| \|A_{\delta,x}\sigma\|_2^2 - \mathbb{E}\|A_{\delta,x}\sigma\|_2^2 \right|. \tag{8}$$

Now, in (2.7) of Bourgain et al. (2015) we instead define a seminorm

$$\|x\|_\delta = \max_{1 \leq i \leq m} \left( \sum_{j \in I^c} \delta_{ij} x_j^2 \right)^{1/2}.$$

Then (2.8) continues to hold, and (2.9) as well as (2.10) continue to hold if the supremum in the left-hand side is replaced with the left-hand side of (8). At the beginning of Theorem 5, we define $U^{(i)}$ to be $U$, but each row $j \in I^c$ is multiplied by $\delta_{ij}$ and each row $j \in I$ is zeroed out. Then we have in the first step of (4.5) that

$$\sum_{j \in I^c} \delta_{ij} \left| \sum_{k=1}^d g_k \langle f_k, e_j \rangle \right|^2 \leq \left\| U^{(i)} g \right\|_2^2,$$

instead of equality. One can verify that the rest of (4.5) goes through. It remains true that $\|\cdot\|_\delta \leq (1/\sqrt{s})\|\cdot\|_2$, and thus (4.6) holds. One can verify that the rest of the proof of Theorem 5 in Bourgain et al. (2015) continues to hold if we replace $\sum_{j=1}^n$ with $\sum_{j \in I^c}$ and $\max_{1 \leq j \leq n}$ with $\max_{j \in I^c}$, noting that

$$\mathbb{E}\sum_{j \in I^c} \delta_{ij} \|P_E e_j\|_2^2 = \frac{s}{m}\sum_{j \in I^c} \langle P_E e_j, e_j \rangle \leq \frac{s}{m}d$$

and

$$\mathbb{E}(U^{(i)})^* U^{(i)} = \sum_{j \in I^c} (\mathbb{E}\delta_{ij}) u_j u_j^* \preceq \frac{1}{m}.$$

Thus, the symmetrization inequalities on

$$\left\| \sum_{j \in I^c} \delta_{ij} \|P_E e_j\|_2^2 \right\|_{L_\delta^p} \quad \text{and} \quad \left\| \sum_{j \in I^c} \delta_{ij} u_j u_j^* \right\|_{L_\delta^p}$$

continue to hold. The result then follows, observing that $\max_{j \in I^c} \|P_E e_j\|^2 \leq \nu$.  □

The subspace embedding guarantee now follows as a corollary.

**Theorem 3.1.** *Let $\nu = \epsilon/d$. Suppose that $m = \Omega((d/\epsilon^2)(\text{polylog}(1/\epsilon) + \log(1/\delta)))$, $\delta \in (0, 1/m)$ and $d = \Omega((1/\epsilon) \, \text{polylog}(1/\epsilon) \log^2(1/\delta))$. Then, there exists a distribution on $S$ with $m + |I|$ rows such that*

$$\Pr\left\{\forall x \in \text{colsp}(A), \left|\|Sx\|_2^2 - \|x\|_2^2\right| > \epsilon \|x\|_2^2\right\} \leq \delta.$$

*Proof.* One can verify that the two conditions in Lemma A.1 are satisfied if

$$m \gtrsim \frac{d}{\epsilon^2}\left(\text{polylog}(\frac{d}{\epsilon}) + \log\frac{1}{\delta}\right),$$

$$d \gtrsim \frac{1}{\epsilon}\left(\log\frac{1}{\delta}\right)\left(\text{polylog}(\frac{d}{\epsilon}) + \log\frac{1}{\delta}\right).$$

The last condition is satisfied if

$$d \gtrsim \frac{1}{\epsilon}\left(\log^2\frac{1}{\delta}\right)\text{polylog}\left(\frac{1}{\epsilon}\right). \qquad \square$$

## B   PROOF OF LEMMA 3.3

*Proof.* On the one hand, since $Q = SAR$ is an orthogonal matrix, we have

$$\|x\|_2 = \|Qx\|_2 = \|SARx\|_2. \tag{9}$$

On the other hand, the assumption implies that

$$\left\|(ARx)^T(ARx) - x^T x\right\|_2 \leq \epsilon \|x\|_2^2,$$

that is,

$$(1 - \epsilon)\|x\|_2^2 \leq \|ARx\|_2^2 \leq (1 + \epsilon)\|x\|_2^2. \tag{10}$$

Combining both (9) and (10) leads to

$$\sqrt{1 - \epsilon}\|SARx\|_2 \leq \|ARx\|_2 \leq \sqrt{1 + \epsilon}\|SARx\|_2, \quad \forall x \in \mathbb{R}^d$$

Equivalently, it can be written as

$$\frac{1}{\sqrt{1 + \epsilon}}\|SAy\|_2 \leq \|Ay\|_2 \leq \frac{1}{\sqrt{1 - \epsilon}}\|SAy\|_2, \quad \forall y \in \mathbb{R}^d.$$

The claimed result follows from the fact that $1/\sqrt{1 + \epsilon} \geq 1 - \epsilon$ and $1/\sqrt{1 - \epsilon} \leq 1 + \epsilon$ whenever $\epsilon \in (0, \frac{\sqrt{5}-1}{2}]$. $\qquad \square$

## C   PROOF OF LEMMA 4.1

Suppose that $AR^{-1} = UW$, where $U \in \mathbb{R}^{n \times d}$ has orthonormal columns, which form an orthonormal basis of the column space of $A$. Since $T$ is a subspace embedding of the column space of $A$ with probability 0.99, it holds for all $x \in \mathbb{R}^d$ that

$$\frac{1}{1 + \eta}\left\|TAR^{-1}x\right\|_2 \leq \left\|AR^{-1}x\right\|_2 \leq \frac{1}{1 - \eta}\left\|TAR^{-1}x\right\|_2.$$

Since

$$\left\|TAR^{-1}x\right\|_2 = \|Qx\|_2 = \|x\|_2$$

and

$$\|Wx\|_2 = \|UWx\|_2 = \left\|AR^{-1}x\right\|_2 \tag{11}$$

we have that

$$\frac{1}{1 + \eta}\|x\|_2 \leq \|Wx\|_2 \leq \frac{1}{1 - \eta}\|x\|_2, \quad x \in \mathbb{R}^d. \tag{12}$$

It is easy to see that

$$Z_1(S) = \min_{x \in \mathbb{S}^{d-1}} \|SUx\|_2 = \min_{y \neq 0} \frac{\|SUWy\|_2}{\|Wy\|_2},$$

and thus,

$$\min_{y \neq 0}(1 - \eta) \frac{\|SUWy\|_2}{\|y\|_2} \leq Z_1(S) \leq \min_{y \neq 0}(1 + \eta) \frac{\|SUWy\|_2}{\|y\|_2}.$$

Recall that $SUW = SAR^{-1}$. We see that

$$(1 - \eta)\sigma_{\min}(SAR^{-1}) \leq Z_1(S) \leq (1 + \eta)\sigma_{\min}(SAR^{-1}).$$

By definition,

$$Z_2(S) = \left\| U^T(S^\top S - I_n)U \right\|_{\mathrm{op}}.$$

It follows from (12) that

$$(1 - \eta)^2 \left\| W^T U^T(S^T S - I_n)UW \right\|_{\mathrm{op}} \leq Z_2(S) \leq (1 + \eta)^2 \left\| W^T U^T(S^T S - I_n)UW \right\|_{\mathrm{op}}.$$

and from (12), (11) and Lemma 5.36 of Vershynin (2012) that

$$\left\| (AR^{-1})^\top(AR^{-1}) - I \right\|_{\mathrm{op}} \leq 3\eta.$$

Since

$$\left\| W^T U^T(S^T S - I_n)UW \right\|_{\mathrm{op}} = \left\| (AR^{-1})^\top(S^T S - I_n)AR^{-1} \right\|_{\mathrm{op}}$$

and

$$\left\| (AR^{-1})^\top S^T SAR^{-1} - I \right\|_{\mathrm{op}} - \left\| (AR^{-1})^\top(AR^{-1}) - I \right\|_{\mathrm{op}}$$
$$\leq \left\| (AR^{-1})^\top(S^T S - I_n)AR^{-1} \right\|_{\mathrm{op}}$$
$$\leq \left\| (AR^{-1})^\top S^T SAR^{-1} - I \right\|_{\mathrm{op}} + \left\| (AR^{-1})^\top(AR^{-1}) - I \right\|_{\mathrm{op}},$$

it follows that

$$(1 - \eta)^2 \left\| (SAR^{-1})^\top SAR^{-1} - I \right\|_{\mathrm{op}} - 3(1 - \eta)^2\eta$$
$$\leq Z_2(S)$$
$$\leq (1 + \eta)^2 \left\| (SAR^{-1})^\top SAR^{-1} - I \right\|_{\mathrm{op}} + 3(1 + \eta)^2\eta.$$

We have so far proved the correctness of the approximation and we shall analyze the runtime below.

Since $S$ and $T$ are sparse, computing $SA$ and $TA$ takes $O(\mathrm{nnz}(A))$ time. The QR decomposition of $TA$, which is a matrix of size $\mathrm{poly}(d/\eta) \times d$, can be computed in $\mathrm{poly}(d/\eta)$ time. The matrix $SAR^{-1}$ can be computed in $\mathrm{poly}(d)$ time. Since it has size $\mathrm{poly}(d/\eta) \times d$, its smallest singular value can be computed in $\mathrm{poly}(d/\eta)$ time. To approximate $Z_2(S)$, we can use the power method to estimate $\left\| (SAR^{-1})^T SAR^{-1} - I \right\|_{op}$ up to a $(1 \pm \eta)$-factor in $O((\mathrm{nnz}(A) + \mathrm{poly}(d/\eta)) \log(1/\eta))$ time.

## D  PROOF OF THEOREM 4.2

In Lemma 4.1, we have with probability at least 0.99 that

$$\frac{\widehat{Z}_2}{\widehat{Z}_1} \geq \frac{\frac{1}{(1+\eta)^2} Z_2(S) - 3\eta}{\frac{1}{1-\eta} Z_1(S)} \geq \frac{1 - \eta}{(1 + \eta)^2} \frac{Z_2(S)}{Z_1(S)} - \frac{3\eta}{Z_1(S)}.$$

When $S$ is random subspace embedding, it holds with probability at least 0.99 that $Z_1(S) \geq 3/4$ and so, by a union bound, it holds with probability at least 0.98 that

$$\frac{\widehat{Z}_2}{\widehat{Z}_1} \geq \frac{1}{(1 + \eta)^4} \frac{Z_2(S)}{Z_1(S)} - 4\eta,$$

or,

$$\frac{Z_2(S)}{Z_1(S)} \leq (1 + \eta)^4 \left( \frac{\widehat{Z}_2}{\widehat{Z}_1} + 4\eta \right).$$

The correctness of our claim then follows from Proposition 1 of Pilanci & Wainwright (2016), together with the fact that $S_2$ is a random subspace embedding. The runtime follows from Lemma 4.1 and Theorem 2.2 of Cormode & Dickens (2019).

## E    PROOF OF THEOREM 5.1

The proof follows a similar argument to that in (van den Brand et al., 2020, Lemma B.1). In van den Brand et al. (2020), it is assumed (in our notation) that $3/4 \leq \sigma_{\min}(AP) \leq \sigma_{\max}(AP) \leq 5/4$ and thus one can set $\eta = 1$ in Algorithm 3 and achieve a linear convergence. The only difference is that here we estimate $\sigma_{\min}(AP)$ and $\sigma_{\max}(AP)$ and set the step size $\eta$ in the gradient descent algorithm accordingly. By standard bounds for gradient descent (see, e.g., p468 of Boyd & Vandenberghe (2004)), with a choice of step size $\eta = 2/(\sigma_{\max}^2(AP) + \sigma_{\min}^2(AP))$, after $O((\sigma_{\max}(AP)/\sigma_{\min}(AP))^2 \log(1/\epsilon))$ iterations, we can find $z_t$ such that
$$\left\| P^\top A^\top AP(z_t - z^*) \right\|_2 \leq \epsilon \left\| P^\top A^\top AP(z_0 - z^*) \right\|_2,$$
where $z^* = \arg\min_z \left\| P^\top A^\top APz - P^\top y \right\|_2$ is the optimal least-squares solution. This establishes Eq. (11) in the proof in van den Brand et al. (2020), and the rest of the proof follows as in there.

We use three subspace embeddings here, $S_1$, $S_2$ and one used in the EIG subroutine. Each subspace embedding uses $O(d^2)$ rows with a constant distortion parameter and a failure probability of $0.01$. The overall failure probability is thus $0.03$.

## F    IHS EXPERIMENTS: MATRIX ESTIMATION WITH NUCLEAR NORM CONSTRAINT

In many applications, for the problem
$$X^* := \arg\min_{X \in \mathbb{R}^{d_1 \times d_2}} \|AX - B\|_F^2 \,,$$
it is reasonable to model the matrix $X^*$ as having low rank. Similar to the $\ell_1$-minimization for compressive sensing, a standard relaxation of the rank constraint is to minimize the nuclear norm of $X$, defined as $\|X\|_* := \sum_{j=1}^{\min\{d_1,d_2\}} \sigma_j(X)$, where $\sigma_j(X)$ is the $j$-th largest singular value of $X$.

Hence, the matrix estimation problem we consider here is
$$X^* := \arg\min_{X \in \mathbb{R}^{d_1 \times d_2}} \|AX - B\|_F^2 \quad \text{such that} \quad \|X\|_* \leq \rho,$$
where $\rho > 0$ is a user-defined radius as a regularization parameter.

We conduct experiments on the following datasets:

- **Tunnel**[4]: The data set is a time series of gas concentrations measured by eight sensors in a wind tunnel. Each $(A, B)$ corresponds to a different data collection trial. $A_i \in \mathbb{R}^{13530 \times 5}$, $B_i \in \mathbb{R}^{13530 \times 6}$, $|(A, B)|_{\text{train}} = 144$, $|(A, B)|_{\text{test}} = 36$. In our nuclear norm constraint, we set $\rho = 10$.

**Experiment Setting**: We choose $m = 7d, 10d$ for the Tunnel dataset. We consider the error $\frac{1}{2} \|AX - B\|_2^2 - \frac{1}{2} \|AX^* - B\|_2^2$. The leverage scores of this dataset are very uniform. Hence, for this experiment we only consider optimizing the values of the non-zero entries.

**Results of Our Experiments**: We plot in a logarithmic scale the mean errors of the two datasets in Figures 5. We can see that when $m = 7d$, the gradient-based sketch, based on the first 6 iterations, has a rate of convergence that is 48% of the random sketch, and when $m = 10d$, the gradient-based sketch has a rate of convergence that is 29% of the random sketch.

**Runtime of Learned Sketch.**    As stated in Section 2, our learned sketch matrices $S$ are all COUNT-SKETCH-type matrices (each column contains a single nonzero entry), the matrix product $SA$ can thus be computed in $O(\text{nnz}(A))$ time and the overall algorithm is expected to be fast. To verify this, we plot in an error-versus-runtime plot for matrix estimation with nuclear norm constraint tasks with $m = 10d$ in Figures 6 (corresponding to the datasets in Figure 5). The runtime consists only of the time for sketching and solving the optimization problem and does not include the time for loading the data. We run the same experiment three times. Each time we take an average over all test data. From the plot we can observe that the learned sketch and COUNT-SKETCH have the fastest runtimes, which are slightly faster than that of the SJLT and significantly faster than that of the Gaussian sketch.

---

[4]https://archive.ics.uci.edu/ml/datasets/Gas+sensor+array+exposed+to+turbulent+gas+mixtures

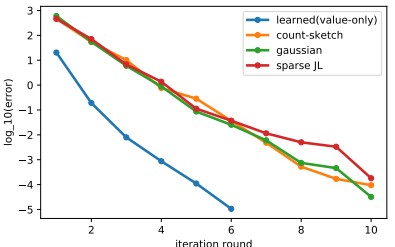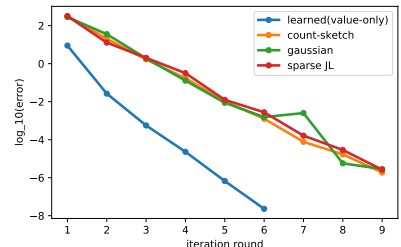

Figure 5: Test error of matrix estimation with nuclear norm constraint on Tunnel dataset

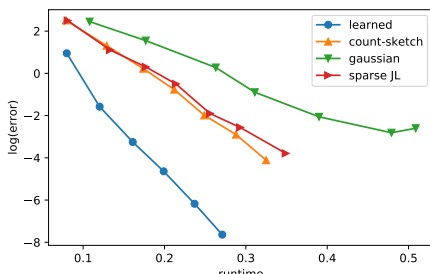

Figure 6: Test error of matrix estimation with nuclear norm constraint on Tunnel dataset

## G   HEAVY LEVERAGE SCORE ROWS DISTRIBUTION OVER THE DATASET

In our experiments, we hypothesize that in real-world data that there may be an underlying pattern which can help us identify the heavy rows. In the Electric dataset, each row of the matrix corresponds to a specific residence and the heavy rows are always concentrated on some specific rows; in the GHG data set, each row corresponds to a specific time point and we can select some specific time points to be a superset of the heavy rows and then select the heavy rows based on their $\ell_2$-norm in this superset.

To exemplify this, we study the heavy leverage score rows distribution over the Electirc dataset. For a row $i \in [370]$, let $f_i$ denote the times that row $i$ is heavy out of 320 training data points from the Electric dataset, where we say row $i$ is heavy if $\ell_i \geq 5d/n$. Below we list all 74 pairs $(i, f_i)$ with $f_i > 0$.

(195,320), (278,320), (361,320), (207,317), (227,285), (240,284), (219,270), (275,232), (156,214), (322,213), (193,196), (190,192), (160,191), (350,181), (63,176), (42,168), (162,148), (356,129), (363,110), (362,105), (338,95), (215,94), (234,93), (289,81), (97,80), (146,70), (102,67), (98,58), (48,57), (349,53), (165,46), (101,41), (352,40), (293,34), (344,29), (268,21), (206,20), (217,20), (327,20), (340,19), (230,18), (359,18), (297,14), (357,14), (161,13), (245,10), (100,8), (85,6), (212,6), (313,6), (129,5), (130,5), (366,5), (103,4), (204,4), (246,4), (306,4), (138,3), (199,3), (222,3), (360,3), (87,2), (154,2), (209,2), (123,1), (189,1), (208,1), (214,1), (221,1), (224,1), (228,1), (309,1), (337,1), (343,1)

Observe that the heavy rows are concentrated on a set of specific row indices. There are only 30 rows $i$ with $f_i \geq 50$. We view this as strong evidence for our hypothesis.

## H   IMPLEMENTATION DETAILS

As we state in Section 3.2, when we fix the positions of the non-zero entries (uniformly chosen in each column or sampling according to the heavy leverage score distribution), we aim to optimize the values by gradient descent mentioned in Algorithm 1. Here the loss function is given in Section 3.2. In our implementation, we use PyTorch (Paszke et al. (2019)), which can compute the gradient automatically (here we can use torch.qr() and torch.svd() to define our loss function). For a more

nuanced loss function, which may be beneficial, one can use the package released in Agrawal et al. (2019), where the authors studied the problem of computing the gradient of functions which involve the solution to certain convex optimization problem.

As mentioned in Section 2, each column of the sketch matrix $S$ has exact one non-zero entry. Hence, the $i$-th coordinate of $p$ can be seen as the non-zero position of the $i$-th column of $S$. In the implementation, to sample $p$ randomly, we can sample a random integer in $\{1, \ldots, m\}$ for each coordinate of $p$. For the heavy rows mentioned in Section 3.1, we can allocate positions $1, \ldots, k$ to the $k$ heavy rows, and for the other rows, we randomly sample an integer in $\{k+1, \ldots, m\}$. We note that once the vector $p$, which contains the information of the nonzero position in each column of $S$, is chosen, it will not be changed during the optimization process in Algorithm 1.

Next, we introduce some parameters for our experiments.

- $bs$: batch size, the number of training samples used in one iteration.
- $lr$: learning rate of the gradient descent(the $\alpha$ in Algorithm 1).
- $iter$: the number of iteration for Algorithm 1.

In our experiments, we set $bs = 20, iter = 1000$ for all dataset. We set $lr = 10$ for the Green House Gas dataset and $lr = 0.1$ for the Electric dataset.

# I    ADDITIONAL EXPERIMENTS FOR LASSO

In this section, we consider the IHS experiments for LASSO on data of a larger size. The experiment setting is the same as that in Section 6. We conduct our experiments on the following dataset:

- **Gas Sensor.**[5]   A chemical detection platform composed of 8 chemoresistive gas sensors was exposed to turbulent gas mixtures generated naturally in a wind tunnel. Each matrix represents the measurements at dense time points during a short time period. $A^i \in \mathbb{R}^{95000 \times 19}, b^i \in \mathbb{R}^{95000 \times 1}$, and $|(A, b)_{\text{train}}| = 30, |(A, b)_{\text{test}}| = 9$. We set $\lambda = 10$.

The results are shown in Table 1. Here we choose $m = 300$. The leverage scores of the rows on this dataset is very uniform hence we choose random positions for the nonzero entries and only optimize the values in the learned sketch. For a matrix of such size, the gaussian sketching matrix is extremely slow, hence, we only consider the Count-Sketch matrix and the Sparse-JL matrix. From the table below we can see that the gradient-based learned sketch has a converge rate that is 74.6% of that of the random sketch.

Table 1: Error of the Sketch Matrix on Gas Senser data

| Iteration | 4 | 5 | 6 | 7 | 8 | 9 | 10 |
|---|---|---|---|---|---|---|---|
| Learnd(value-only) | 428.13 | 23.72 | 1.72 | 0.092 | 0.0060 | 0.00036 | $1.8 \cdot 10^{-5}$ |
| Count-Sketch | 1864.66 | 122.26 | 12.30 | 1.46 | 0.074 | 0.013 | 0.00036 |
| Sparse-JL | 1897.32 | 188.31 | 9.60 | 1.30 | 0.16 | 0.0098 | 0.00048 |

---

[5]https://archive.ics.uci.edu/ml/datasets/Gas+sensor+array+temperature+modulation

