# OpenReview forum: "Learning-Augmented Sketches for Hessians "
_ICLR.cc/2022/Conference — ICLR 2022 Submitted_

### Official Review · Reviewer_SAjC · 2021-10-19

**Correctness:** 4
**Technical Novelty And Significance:** 2
**Empirical Novelty And Significance:** 3
**Recommendation:** 5
**Confidence:** 3

**Main Review:**

My background is more in theoretical matrix algorithms, so to me, the idea of adaptively generating sketches based on leverage scores is somehow more fundamental than sketching. The authors do do a good job addressing this related literature, but their results are mostly direct consequences of matrix concentration bounds.

The experimental results are interesting: I'm not aware of such evaluations of test errors in previous works. They also point to significant gains of these methods over oblivious sketching based methods. On the other hand, these studies mostly take place on small to moderate sized data where the performance gains from sketching is unclear. So I feel they serve mostly as proof of concept of the utility of such sketching, and perhaps also point out that different formulations/objectives are necessary for giving better test errors.

**Summary Of The Paper:**

This paper studies matrix sketching algorithms where the sketching matrix is `learned', or generated adaptively. Theoretically, this is done by favoring more the rows of A with high leverage scores, while practically this is done by associating that set with rows of large norms. It evaluates this algorithm on both L_1 and L_2 regression, and observes improved test errors.

**Summary Of The Review:**

I feel this paper is well thought out, and the experimental results are highly interesting. However, I also feel that the studies taken here (adaptively select rows) can be taken one step deeper: in particular, it would be very interesting to see theoretical analyses of why such learned sketching give improved results (over oblivious sketches).

---

> ### Author Response · Authors · 2021-11-15
> **Responses**
>
> We thank the reviewer for his/her careful comments. We have added an experiment with a larger size matrix in Appendix I. The following are our responses to the comments.
>
> In our theorem, we show that if we can find a superset that contains the rows of heavy leverage score without obtaining the leverage scores precisely, then we can also reduce the number of rows in a Count-Sketch-type sketching matrix from $O(d^2/\epsilon^2)$ to $O(d/\epsilon^2)$ if the data does not have many rows with heavy leverage score (we think this is common for real-world data). Additionally, in our empirical results, we show it is possible to learn such row indices for real-world data (for the Electric data we do not compute the norm of the rows but compute the statistics over the training set of the frequency of each row having a large leverage score).
>
> For learning the non-zero values, we also give a theorem saying that if the test data can achieve a small loss under the loss objective we designed, then the algorithm can attain a fast convergence rate. Under the assumption that the training set and testing test come from a similar distribution, it is reasonable to believe that a small loss on the training set means also a small loss on the testing set.
>
> We agree with the reviewer's point that the study of adaptive row selection can be taken deeper, and we see it as an interesting future direction.

---

### Official Review · Reviewer_kP9y · 2021-10-23

**Correctness:** 4
**Technical Novelty And Significance:** 2
**Empirical Novelty And Significance:** 2
**Recommendation:** 5
**Confidence:** 3

**Main Review:**

# Strengths

S1. Learning of sketch matrices is an interesting twist to the more standard random sketches out there.

S2. The theory for the learned heavy row sketch is a nice attempt at providing justification for why the proposed learning method might improve results.

# Weaknesses

W1. It is unclear how useful these learned sketches actually are. In applications, it seems like the learned sketches could easily break. To put it in machine learning terms, the heavy row sketch uses row indices as features to predict whether or not rows are heavy. This is somewhat akin to classifying images on a social networking site based on the ID of the user who uploaded them: if the user with that ID uploaded mostly cat pictures in the training data, then classify the newly uploaded picture as a cat picture. This means that if the rows of a design matrix in the testing data are permuted, then the learned heavy row sketch isn’t useful anymore. This could for example happen in the electricity dataset if two residents switch homes. It therefore seems like, in practice, you would need to retrain the model frequently to ensure that the leverage score estimates remain accurate.

W2. The datasets in the experiments are very small scale, and can be solved very easily using deterministic methods. It would have been more interesting to see the performance on larger scale datasets where these kind of techniques would potentially be more useful.

W3. The improvements that the learned sketches yield don't seem that substantial in some of the experiments. For example, in Figures 1 and 4-6, it seems like they just save a few iterations compared to doing standard CountSketch.

W4. Some parts of the paper are unclear; see questions below.

# Questions

Q1. How is line 10 in Alg. 2 computed?

Q2. Below Theorem 4.2, you discuss how “a better subspace embedding can lead to a faster convergence.” But it’s not clear from the discussion why this is the case. In particular, it’s not clear why the discussion around improved accuracy guarantees would lead to improved convergence rates. What am I missing?

Q3. Out of curiosity, how do you implement the various CountSketch matrices? Do you form sparse matrices in Python with the relevant nonzeros and multiply, or do you do something more refined to make the computation faster?

Q4. The datasets are poorly introduced. What does $b$ represent in the Electric and GHG datasets? How is the data in $A$ in the GHG dataset organized---is time along the rows with different columns corresponding to different measurement sites?

Q5. Also, does |$(A,b)$_train|=400 and |$(A,b)$_test|=100  mean that you have a total of 500 pairs of design matrices and corresponding vectors $b$, and that you use 400 for training and the rest for testing? If yes, are all the reported results the average results over all pairs $(A,b)$ in the testing data?

Q6. It’s not clear why some methods are not given in some plots. Why is “learned(combined)” not plotted in Figure 1 when it appears in Figure 2? Why are “learned(value-only)” and “learned(combined)” not included in Figure 3? Why are “learned(value-only)”, “learned(combined)” and CountSketch not included in Figure 4?

Q7. What is the difference between the three plots in Figure 1? Please explain in the caption. Also, the y-axis labels are cut off and not showing properly.

Q8. In Section 6.2, why are you using Newton’s method to solve a least squares problem? Is there any benefit to doing that rather than just doing a single sketch-and-solve on the least squares problem itself?

Q9. In the last paragraph before the conclusion, you say that you “only run one iteration” for each subroutine. What does that mean? It looks like you do multiple iterations in Figure 4?

Q10. One of the questions you mention in the introduction is “(2) should we apply the same learned sketch in each iteration, or learn it in the next iteration by training on a data set involving previously learned sketches from prior iterations?” You then say that you will answer this question, but I didn’t see this discussed in the paper. You briefly mention it in the paragraph “Training” in Section 6.2, but I didn’t see it properly addressed anywhere. Did I miss something?

# Other minor things

- In footnote on page 3, “inut” should be “input”.

- The “Conclusion” heading is too big for a paragraph heading. You should put it as a proper section heading.

- You should specify the unit used in the “runtime” label on the x-axis of Figure 6. I’m assuming it’s seconds?


**Summary Of The Paper:**

This paper considers how learned CountSketches can be used in a variety of optimization tasks. The authors propose a method for predicting rows with high leverage score based on the training data. These rows are then sampled deterministically alongside either a standard CountSketch, or the learned CountSketch introduced by Indyk et al. (2019). Some novel theory is presented for this new sketch.

**Summary Of The Review:**

The core idea of learning sketch matrices is interesting. The theory that provides a potential justification for why learned heavy row sketching might improve on the oblivious CountSketch is welcome. However, the “learned oracle” for the heavy row sketch seems very easy to break: Simply permuting rows would mean that it has to be relearned. Overall, it’s not clear how useful the learned sketch would be in practice for that reason. The experiments are all done on very small problems that don’t need sketching. Parts of the paper are also unclear as outlined above. For these reasons, I think it’s below the acceptance threshold.

---

> ### Author Response · Authors · 2021-11-15
> **Responses to Weaknesses**
>
> We thank the reviewer for his/her careful comments. The following are our responses.
>
> > 1. It is unclear how useful these learned sketches actually are.
>
> To address the concern that a permutation may break the sketch, we can measure the similarity between vectors, that is, after processing the training data, we can instead test similarity on the rows of the test matrix and use this to select the heavy rows, rather than an index which may simply be permuted. To illustrate this method, we use the following example on the Electric dataset, using locality sensitive hashing.
>
> After processing the training data, we obtain a set $I$ of indices of heavy rows. For each $i\in I$, we pick $q = 3$ independent standard Gaussian vectors $g_1, g_2, g_3\in \mathbb{R}^d$, and compute $f(r_i) = (g_1^T r_i, g_2^T r_i, g_3^T r_i) \in \mathbb{R}^3$, where $r_i$ takes an average of the $i$-th rows over all training sets. Let $A$ be the test matrix. For each $i\in I$, let $j_i = \operatorname{argmin}_{j} \Vert f(A_j) - f(r_i)\Vert_2$. We take the $j_i$-th row to be a heavy row in our learned sketch. This method only needs an additional $O(1)$ passes over the entries of $A$ and hence, the extra time cost is negligible. To test the performance of the method, we randomly pick a matrix from the test set and permute its rows. The result shows that when $k$ is small, we can roughly recover $70\%$ of the top-$k$ heavy rows, and we list below the regression error using the learned Count-Sketch matrix generated this way, where we set $m = 90$ and $k = 0.3m = 27$. We can see that the learned method still obtains a  significant improvement.
>
>                      1        2       3       4         5         6          7
>     Learned         3.423   0.196  0.00069  1.576e-5  1.805e-6  4.060e-8   1.360e-9
>     Count-Sketch   53.183   2.282  1.040    0.0409    0.00354   0.000113   1.777e-5
>     Gaussian       60.436   2.601  0.622    0.0587    0.00865   0.00171    0.000172
>     SJLT           74.071   4.482  0.647    0.0619    0.00926   0.000925   0.00026
>
> Furthermore, as suggested in the previous result [2] for the frequency estimation problem, in practice there could be more aspects that can be exploited in learning algorithms, such as the number of IP addresses or the number of characters in words. The result [2] shows that based on these features, machine learning methods can achieve a good performance. Hence, we believe that the proposed learning-based method in our paper will be more powerful for real-world data.
>
> > 2. The improvements that the learned sketches yield don't seem that substantial in some of the experiments.
>
> We respectfully disagree with the view that the improvement is not substantial. As we analyzed, in Figure 1 and Figures 5-6, the learned-sketch (value) can achieve a convergence rate that is 56%, 48% and 29% of the random sketch, which means half the time or even less to achieve the same error. For Figure 4, we note that the algorithm here has an inner iteration and an outer iteration. The results of the inner iteration are shown in Figure 3, from which we can see the learned sketch can at least give a speedup of a factor of 3 if we want the inner iteration to achieve a small error.
>
> [2]: Chen-Yu Hsu, Piotr Indyk, Dina Katabi and Ali Vakilian. Learning-Based Frequency Estimation Algorithms. ICLR 2019.
>
> > 3. The datasets in the experiments are very small scale.
>
> We have added an experiment with a larger size matrix in Appendix I, from which we can see the benefits of the sketch method.

---

> > ### Comment · Reviewer_kP9y · 2021-11-16
> > **Thank you for the response**
> >
> > Thank you for the detailed response to my review.
> >
> > Although the added experiment in Appendix I is larger, it is still quite small scale. Solving a linear system of this size on my desktop computer with the lasso function in Matlab takes about 0.5 seconds, so it is still not large enough for sketching to make sense.
> >
> > The idea to use a few Gaussian test vectors to help find heavy rows even if they have been permuted is interesting. However, the paper would have to be substantially revised to incorporate this idea.
> >
> > Since the only update made to the paper is the addition of Appendix I, I'm leaving my score as it is.

---

> ### Author Response · Authors · 2021-11-15
> **Responses to Questions**
>
> Q1. It is explained at the end of Appendix C that one can use the power method to estimate $\Vert (SAR^{-1})^T SAR^{-1}-I\Vert_{\mathrm{op}}$ up to a $(1\pm\eta)$-factor in $O((\operatorname{nnz}(A)+\operatorname{poly}(d/\eta))\log(1/\eta))$ time.
>
> Q2. The convergence rate depends on the quantity $\rho = Z_2/Z_1$, where $Z_1$ and $Z_2$ depend on the accuracy parameter of the subspace embedding property. It was shown in Theorem 22 of (Pilanci and Wainwright, 2016) that $t$ rounds of the Iterative Hessian Sketch yields an $x_t$ such that $\Vert A(x_t-x^\ast)\Vert_2 \leq \rho^t \Vert Ax^\ast\Vert_2$. Here a smaller distortion of the subspace embedding property implies a smaller $\rho$, and thus improves the convergence rate.
>
> Q3: We give a more detailed description in Appendix~G about how to form the Count-Sketch matrix. After forming the sketching matrix $S$, we apply $S$ to $A$, obtaining $SA$ in $O(\operatorname{nnz}(A))$ time.
>
> Q4: The task in our experiment for the Electric dataset is to use the previous information of the usage from the user to predict future usage. Yes, for the GHG data, the time along the rows with different columns corresponds to different measurement sites.
>
> Q5: Yes, all the reported results a	re the average results over all pairs in the test data.
>
> Q6: For the Electric dataset, the non-zero positions of the learned sketch matrix are the same for all test data. Hence, we can do the combined method, that is, first learn the positions, then learn the values. For the GHG dataset, the positions are slightly different across the test data (as we mentioned in the experiments section), and hence we cannot use the combined method here. For Figures 3 and 4, we observe that the learned sketch (position) can achieve a tiny error. Hence, we do not report the result for the learned sketch (value). Furthermore, we mentioned on Page 9 that the random count-sketch matrix causes a terrible divergence of the subroutine in this task and is thus omitted in the plots (this can also be seen as an advantage of the learned matrix).
>
> Q7: The difference across the three plots is the sketch size (from $6d$ to $10d$).
>
> Q8: Consider the dependence of $\epsilon$ in the sketch size for a $(1 \pm \epsilon)$-solution. A single sketch-and-solve method needs $O(\frac{1}{\epsilon})$ in the sketch size while an iterative method can achieve $O(\log \frac{1}{\epsilon})$.
>
> Q9: As we mentioned above, the algorithm of Figures 3 and 4 has an inner iteration (the iterative method to solve Eq. (3)) and an outer iteration (Eq. (3)). The results of the inner iteration are shown in Figure 3 and those of the outer iteration are shown in Figure 4, where we fix all inner steps to one iteration only. The intuition of Figure 4 is to show that the accuracy of the inner iteration will affect the outer iteration.
>
> Q10: One of the questions mentioned in the introduction.
>
> We agree with the reviewer that we do not discuss it in detail.
>
> For this question, there are three naive ways: (i) training a single sketch matrix for all iterations, (ii) training $k$ sketching matrices (one from each round) using a global loss function on $k$ matrices altogether, or (iii) training $k$ sketching matrices sequentially, that is, learning the sketching matrix for the $i$-th iteration using the local loss function for the $i$-th iteration, and then using the learned matrix in the $i$-th iteration to generate the training data for the $(i+1)$-st iteration.
>
> However, after our exploration, all three methods failed.
> For (i), the theoretical analysis of IHS needs the sketch matrices across each iteration to be independent, and we observe that using the same sketch matrix may cause the solution to diverge. For (ii), the training is computationally prohibitive because the number of variables becomes $k$ times larger; And for (iii), the empirical results show improvement for the first iteration only, because the training data becomes farther away from the test data in later iterations. The core problem here is that the method proposed in previous work focuses on minimizing the loss function for their specific problem, which is difficult to extend to iterative methods.
>
> In our work, to address these issues, we notice that the desired property for the sketching matrix is the subspace embedding property. To optimize this, we propose a new way to learn the positions and employ a new loss function,  focusing on the subspace embedding property. Our empirical results show that learned sketches can achieve a much better performance than random sketches.
>
> We think it is a good question from the reviewer and we will add more explanation about this in the appendix.

---

### Official Review · Reviewer_zU27 · 2021-10-27

**Correctness:** 4
**Technical Novelty And Significance:** 2
**Empirical Novelty And Significance:** 2
**Recommendation:** 5
**Confidence:** 3

**Main Review:**

## Strengths:
The paper is well written and gives a nice background of existing literature with strong motivation. The technical detail is of good quality and the numerical experiments provided show that the method performs well on selected data sets when comparing number of iterations required. The idea is simple and intuitively appealing. The problems addressed, specifically least squares, are ubiquitous and finding efficient ways to solve high dimensional problems is of perennial interest. In addition, error/complexity bounds are provided when used in Iterative Hessian Sketch and Fast Regression.


## Weaknesses
The paper claims to provide a framework for learned sketching that applies to a large number of problems in convex optimization but still focuses on least squares problems. It seems that the primary contribution of this paper is to apply learned sketches for least squares problems as introduced in works by Liu et al (_On learned sketches for randomized numerical linear algebra [2020]_ and _Learning the positions in countSketch [2020]_) to iterative hessian sketch and fast regression. Although theoretical bounds are provided, the contribution is more limited than initially stated.



## Questions and comments:
1. Is it common that the Hessian can be decomposed as $A^T A$ with $A \in \mathbb{R}^{n \times d}$ and $n \gg d$ for other convex problems? This is common for LS but is it often observed elsewhere that makes it more useful for non LS problems.
2. For learning the locations of the non-zero entries as discussed in 3.1, how is the oracle trained to predict these heavy rows? Are the number of occurrences of heavy rows tabulated and the the $k$ most common rows to have leverage scores over some threshold selected? More detail on this front would be helpful.
3. Although the sketches can be learned offline and finished "within 5 minutes", it is unclear how the sketch training time compares with the time to solve the problem. A comparison of training time and time to convergence would be helpful to evaluate its merit.



## Minor comments
- $p \in \mathbb{N}^n$ rather than being a real vector
- The notation $|(A_i, b_i)|$ isn't entirely clear. It clear indicates the sixe

**Summary Of The Paper:**

In this paper, the authors extend a line of work focused on sketching the Hessian for convex problems to help accelerate second order optimization methods. In particular, they present an algorithm for learning weights of a sketching matrix (with one non-zero entry per columns chosen uniformly) by using gradient descent; the Hessians used for training are treated as draws from a distribution of Hessians. They also discuss how leverage scores can be used to improve convergence rates by ensuring that "heavy" rows are sampled with probability 1.

The authors show empirically that the learned sketches improve convergence rates and reduce the number of iterations for several problems. They also provide theoretical results stating a reduction in rows required in the sketch when heavy leverage scores rows are known and error/time complexity bounds for the Hessian sketch/regression problems.

**Summary Of The Review:**

Since the paper primarily focuses on the application of a learned sketch to an iteratively solved LS problem (both components well established elsewhere) the contribution seems marginal. I believe the paper is marginally below the acceptance threshold as is. The paper can be improved by addressing questions and comments above, in particular, evidence that the method is applicable to more general convex problems and a more comprehensive comparison for total time to solve compared to naive sketching.

---

> ### Author Response · Authors · 2021-11-15
> **Responses**
>
> We thank the reviewer for his/her careful comments. We have added an experiment with a larger size matrix in Appendix I. The following are our responses.
>
> > 1. The framework was proposed in Liu et al.
>
> We agree with the reviewer's point that the framework is mentioned by Liu et al. However, both methods (one optimizes the positions and the other the values) proposed by Liu et al. cannot be applied in our settings. Take the learned IHS for example: here we need a different sketching matrix in each iteration (otherwise the error may not converge). Suppose that we are to train the matrices for $k$ iterations. If we were to follow the method in Liu et al, we could (i) train $k$ sketching matrices (one from each round) using a global loss function on $k$ matrices altogether, or (ii) train $k$ sketching matrices sequentially, that is, learn the sketching matrix for the $i$-th iteration using the local loss function for the $i$-th iteration, and then use the learned matrix in the $i$-th iteration to generate the training data for the $(i+1)$-st iteration. However, in (i), the training is computationally prohibitive because the number of variables becomes $k$ times larger; in (ii), the empirical results show improvement for the first iteration only, because the training data becomes farther away from the test data in later iterations. The core problem here is that the method proposed in Liu et al. focuses on minimizing the loss function for their specific problem, which is difficult to extend to iterative methods.
>
> However, in our work, the desired property for the sketching matrix is the subspace embedding property. To optimize this, we employ a new loss function focusing on the subspace embedding property. The empirical results show that the learned sketches can achieve a much better performance than random sketches.
>
> In fact, in an independent work in~[1] (NeurIPS 2021), the authors also observed that the previous loss function is not good enough and proposed a new one, catering to the property desirable for low-rank approximation.
>
> > 2. Is it common that the Hessian can be decomposed as $A^T A$ with $A \in \mathbb{R}^{n \times d}$ and $n \gg d$ for other convex problems? This is common for LS but is it often observed elsewhere that makes it more useful for non LS problems.
>
> It is indeed true that for some problems, one cannot obtain an explicit form of $A$, as needed to apply a sketching matrix, though our framework applies to the widely studied empirical risk minimization problem $\min_{w\in \mathbb{R}^d} \{\frac{1}{n}\sum_{i=1}^n f(\langle w,\Phi(x_i)\rangle,y_i)\}$, where $A\in \mathbb{R}^{n\times d}$ ($n\gg d$) can be constructed easily by letting its $i$-th row $A_i$ be equal to  $\sqrt{\frac{1}{n}f''(\langle w,\Phi(x_i)\rangle,y_i)}\Phi(x_i)^T$.
>
> > 3. For learning the locations of the non-zero entries as discussed in 3.1, how is the oracle trained to predict these heavy rows?
>
> Our hypothesis is that many real-world data sets may have an underlying pattern which can help us identify the heavy rows. Our experiments exemplify this. For instance, in the Electric dataset, each row of the matrix corresponds to a specific residence and the heavy rows are always concentrated on a small subset of rows (we give a more detailed explanation in Appendix~G); in the GHG data set, each row corresponds to a specific time point and we can select some specific time points to be a superset of the heavy rows and then select the heavy rows based on their $\ell_2$-norms.
>
> > 4. It is unclear how the sketch training time compares with the time to solve the problem.
>
> We included a time-versus-error plot in Appendix F. Note that the learned matrix $S$ is trained offline only once using the training data. Hence, there is no additional cost during testing.
>
> [1] Piotr Indyk, Tal Wagner and David Woodruff. Few-Shot Data-Driven Algorithms for Low Rank Approximation. NeurIPS 2021

---

> > ### Comment · Reviewer_zU27 · 2021-11-22
> > **Thank you for the response**
> >
> > Thank you to the authors for their response to this review and others, it helped clarify several issues.
> >
> > I am still uncertain how the runtime plot in appendix F addresses my concern above. There is a cost to train the sketch and a cost to solve the problem whether it's solved exactly, or after sketching (naive or learned). In practice, one must first train the sketch then solve the sketched problem. The time required to train a sketch can't be ignored even if training is only performed once. If it takes 5 minutes to train and 0.5 seconds to solve the cost of learning a sketch would be prohibitively high. Although the solution might be more accurate than an oblivious sketch, there is an additional cost. Am I missing something?
> >
> > Several of the concerns raised by other reviewers are valid, in particular, issues with permuting data and using more realistic examples. To get the paper in conference ready form would require substantial reworking. For that reason, I am leaving my review unchanged.

---

> > > ### Author Response · Authors · 2021-11-29
> > > **Responses**
> > >
> > > We thank the reviewer for the reply.
> > >
> > > It's true that the pre-training over data in the training set will incur additional computation. However, as mentioned in [1], in many applied scenarios of the sketching method, we need to process the streams of data (video, data logs, customer activity etc) by executing the same algorithm on an hourly, daily or weekly basis. Hence we can finish the offline training tasks before the new data comes. It is acceptable if the training can be finished in minutes (or even longer) in the work of learning-based algorithms in recent years, see, e.g., [1], [2], and [3].
> > >
> > > [1] Piotr Indyk, Ali Vakilian, Yang Yuan. Learning-Based Low-Rank Approximations. NeurIPS 2020
> > >
> > > [2] Chen-Yu Hsu, Piotr Indyk, Dina Katabi and Ali Vakilian. Learning-Based Frequency Estimation Algorithms. ICLR2019
> > >
> > > [3] Simin Liu, Tianrui Liu, Ali Vakilian, Yulin Wan, David P. Woodruff. Learning the Positions in CountSketch

---

### Official Review · Reviewer_tMi8 · 2021-11-02

**Correctness:** 3
**Technical Novelty And Significance:** 3
**Empirical Novelty And Significance:** 3
**Recommendation:** 5
**Confidence:** 3

**Main Review:**

**Pros:**

- This paper shows provable speed-ups on the second-order methods introduced in previous works, and further demonstrates with experiments.
- The previous works are properly surveyed and discussed in the context of the proposed methods.


**Cons, and questions to authors:**

Despite the above contributions, the writing is ambiguous and does not properly draw relationships among different parts, especially among Section 2, 3 and 4. I would increase my score if the authors could properly address the doubts.

- How does COUNT-SKETCH introduced in Section 3.1 relate to the methods proposed in Section 4 and 5? It seems both Section 4 and 5 rely on the QR method in Section 3.2; the results in 3.1 are more of standard leverage score sampling results and are irrelevant to the proposed approaches.
- How can the matrix $T$ be quickly chosen in Algorithm 2 Line 7, and what is the relationship between $T$ and $S_i$, and $T$ and $A$? It is not clear from the writing why such a $T$ is readily available.
- Cost of each iteration: at the bottom of Page 7, the authors claim that “no additional computational cost is incurred in generating $S$ other than solving the iteration step using COUNT-SKETCH”, and Guassian matrices and sparse JL transforms "will be considerably slower". However, the iteration step itself requires running Algorithm 2 or 3, each of which includes QR on the original $A$ matrix (Algorithm 2 Line 8, Algorithm 3 Line 2), which are also expensive. I would suggest the authors compare per-iteration complexities of the approaches shown in Figure 1. Besides, it is not clear how Algorithm 2 Line 10 is implemented.
- Section 3.2 at the bottom of Page 5 says “We found empirically that not squaring this loss function works better than squaring it”. Why does (not-)squaring matter if $S$ is solved by just minimizing the loss $\mathcal{L}(S, A_i)$ itself?


**And some minor issues:**

- The notations at the bottom of Page 5 and in Lemma 3.3 at the beginning of Page 6 are inconsistent: Page 5 says $SA_i = Q_i R_i^{-1}$, whereas Page 6 says $SA = QR$. Also, it would be better if the authors could expand the $\mathcal{L}(S, A_i)$ term at the bottom of Page 5 to explicitly show how S goes into the loss.
- What are the $A_i$, $b_i$s in Section 6.1 below Equation 7, are they just $A$ and $b$ (the features and labels of the corresponding datasets)?
- It may be nice to show these numerics: the convergence of first-order methods in FIgure 1-4 as a baseline, and how the spectrum of $A$ looks like.

**Summary Of The Paper:**

This paper shows sketching methods to speed up iterative Hessian sketch (IHS) and Hessian regression. With decently accurate subspaces learned from QR decomposition, IHS and Hessian regression achieve faster convergence than data oblivious methods.

**Summary Of The Review:**

As mentioned above, despite the contributions, the writing (if not the method itself) is often confusing. I would suggest the authors address these ambiguities in both the author response and the revised version.

---

> ### Author Response · Authors · 2021-11-15
> **Responses**
>
> We thank the reviewer for his/her careful comments. We have added an experiment with a larger size matrix in Appendix I. The following are our responses.
>
> > 1. How does COUNT-SKETCH introduced in Section 3.1 relate to the methods proposed in Section 4 and 5?
>
> In the algorithms of Sections 4 and 5, we test whether the trained matrix is better than a random sketching matrix and give theoretical guarantees on what we can gain if we can have a good training error. It is a coincidence that both Sections 3.2 and 4 invoke the QR decomposition. In Section 3.2, the QR decomposition is used in the definition of the loss function for training the sketch matrix, while in Section 4 it is used in testing whether the trained matrix is better. We will add more explanations on this point.
>
> > 2. How can the matrix $T$ be quickly chosen in Algorithm 2 Line 7, and what is the relationship between $T$ and $S_i$, and $T$ and $A$? It is not clear from the writing why such a $T$ is readily available.
>
> The matrix $T$ is chosen independently with $S$ and can be chosen to be a COUNT-SKETCH matrix in which the nonzero positions are pairwise independent and the signs 4-wise independent. The subspace embedding property still holds with $O(d^2/\epsilon^2)$ rows of $T$ (Nelson and Nguyen, 2013). We will add the explanation about COUNT-SKETCH with limited independence. We do not need to generate the matrix $T$ explicitly as we only need to ensure that the sketched version of $A$, namely $TA$, can be computed efficiently. For each nonzero entry of $A$, it is multiplied by exactly one nonzero entry of $T$, which we can calculate efficiently in $O(1)$ time. Thus the total time to compute $TA$ remains $O(\operatorname{nnz}(A))$.
>
> Here we want $T$ to preserve the column space of $A$. Since the COUNT-SKETCH matrix gives an oblivious subspace embedding, that is, it preserves a subspace with constant probability for any subspace, $T$ can be constructed independently of $A$. In fact, a point of our paper is that the dimension of the subspace embedding matrix can be reduced by allowing the construction to depend on $A$.
>
> > 3. Cost of each iteration.
>
> In our experiments on learned sketches, we use the learned sketch matrix in all iterations. Hence, there is no additional cost here. We included a time-versus-error plot in Appendix~F for the matrix estimation problem with a nuclear norm constraint.
>
> Also, in Algorithms 2 and 3, the QR decomposition is applied after multiplying a sketching matrix $T$, which has $\operatorname{poly}(d/\epsilon)\ll n$ rows, and hence it will be much faster than doing a QR decomposition on $TA$ than on the original matrix $A$.
>
> > 4. Besides, it is not clear how Algorithm 2 Line 10 is implemented.
>
> It is explained at the end of Appendix C that one can use the power method to estimate $\Vert (SAR^{-1})^T SAR^{-1}-I\Vert_{\mathrm{op}}$ up to a $(1\pm\eta)$-factor in $O((\operatorname{nnz}(A)+\operatorname{poly}(d/\eta))\log(1/\eta))$ time.
>
> > 5. Why does (not-)squaring matter if $S$ is solved by just minimizing the loss itself?
>
> We use SGD during the optimization process. That is, in every iteration, we select a subset from the training set and then compute the loss over the subset. Suppose that the set we select is $\{A_1, \dots, A_k\}$. Then the loss function of the squared version is $\sum_{i=1}^k \mathcal{L}(S,A_i)^2$, while the loss function of the non-squared version is $\sum_{i=1}^k \mathcal{L}(S,A_i)$, which is less sensitive to outliers.

---

> > ### Comment · Reviewer_tMi8 · 2021-11-29
> > **Further questions**
> >
> > Thanks for the detailed response. Here are some remaining questions:
> >
> >
> > 1.
> > > We do not need to generate the matrix $T$ explicitly as we only need to ensure that the sketched version of $A$, namely $TA$, can be computed efficiently.
> >
> > Why is that? If we do not need an explicit $T$, how would Line 7 and 8 of Algorithm 2 be implemented?
> >
> > 2. Thanks for the explanation on the per-iteration cost. Now I have the same doubt as Reviewer kP9y: would the additional cost of sketching pay off, compared to having no sketches at all? This is because although the per-iteration cost to solve Problems (5) and (6) are made lower in the proposed methods, Problems (5) and (6) are themselves subroutines of the original problems you are trying to solve: for example, an unconstrained convex optimization problem. Are there empirical results that compare the proposed methods with first-order methods on the datasets used in Section 6 (like, the accuracy achieved within a given time)?

---

> > > ### Author Response · Authors · 2021-11-30
> > > **Responses**
> > >
> > > > 1. Why is that? If we do not need an explicit , how would Line 7 and 8 of Algorithm 2 be implemented?
> > >
> > > It is not necessary to generate or store the entire matrix $T$ explicitly since Line 8 depends only on $TA$. The algorithm works as long as we can generate $TA$ efficiently. For example, we can take $T$ to be a Count-Sketch matrix of $m = O(d^2/\eta^2)$ rows, which is characterized by a hash function $h:[n]\to [m]$ and $n$ random signs $\sigma_1,\dots,\sigma_n$. We first initialize an $m\times d$ zero matrix $F$. Whenever we see a nonzero entry at $(i,j)$-th position of the matrix $A$, we can let $F_{h(i), j} \gets F_{h(i), j} + \sigma_i A_{ij}$. After processing all nonzero entries of $A$, the matrix $F$ is exactly $TA$. Furthermore, the hash function $h$ can be chosen to be pairwise independent and the random signs $\{\sigma_i\}$ are $4$-wise independent. For a given $i$, we can calculate $h(i)$ and $\sigma_i$ in constant time. Hence, $TA$ can be computed in $O(nnz(A))$ time and there is no need to first generate an $m\times n$ matrix $T$.
> > >
> > > > 2. Thanks for the explanation on the per-iteration cost. Now I have the same doubt as Reviewer kP9y: would the additional cost of sketching pay off, compared to having no sketches at all? This is because although the per-iteration cost to solve Problems (5) and (6) are made lower in the proposed methods, Problems (5) and (6) are themselves subroutines of the original problems you are trying to solve: for example, an unconstrained convex optimization problem. Are there empirical results that compare the proposed methods with first-order methods on the datasets used in Section 6 (like, the accuracy achieved within a given time)?
> > >
> > > Below we list the time vs error table for the least-squares problem, using the standard gradient descent method. We test the error on the Electric dataset and take an average over 80 testing matrices. Here we set the learning rate $\alpha = 10^{-4}$ (a larger learning rate will cause $x$ to diverge).
> > >
> > >     iteration       10          100         500        1000         10000
> > >     error        69.35        26.45        9.42        5.81         0.795
> > >     time(sec)   0.0003      0.00263      0.0148       0.033        0.2907
> > >
> > > From the table, we can see that the gradient descent method will need about $10^4$ rounds to achieve an error of less than $1$. While the second-order method in our paper will need approximately 8 rounds, which takes about 0.023 seconds.
> > >
> > > As we replied to Reviewer zU27, it is true that the pre-training over data in the training set will incur additional computation. However, as mentioned in [1], in many applied scenarios of the sketching method, we need to process the streams of data (video, data logs, customer activities, etc.) by executing the same algorithm on an hourly, daily or weekly basis. Hence we can finish the offline training tasks before the new data comes. It is acceptable if the training can be finished in minutes (or even longer) in the work of learning-based algorithms in recent years, see, e.g., [1], [2], and [3].
> > >
> > > We also study the use of the sketch matrix in first-order methods. Particularly, let $QR^{-1} = SA$ be the QR-decomposition for $SA$, where $S$ is a sketch matrix. We use $R$ as an (approximate) pre-conditioner and use the gradient descent to solve the problem $\min \Vert ARx - b\Vert_2^2$. Here $A$ is $370\times 9$ and we set $S$ to have $90$ rows. The result is shown in the following table, where the time includes the time to compute $R$. We can see that if we use a learned sketch matrix, the error converges very fast when we set the learning rate to be 1 and 0.1, while the classical Count-Sketch matrix will lead to divergence.
> > >
> > >         iteration                         1             10            100         500
> > >     error(learned, lr = 1)             2.73         1.5e-7
> > >     error(learned, lr = 0.1)           4045            605        4.04e-6
> > >     error(learned, lr = 0.01)          4897           4085            667       0.217
> > >     error(random, lr = 1 or 0.1)                    diverge
> > >     error(random, lr = 0.01)           4881           3790            685       1.52
> > >     time                            0.00048        0.00068         0.0029       0.013
> > >
> > > [1] Piotr Indyk, Ali Vakilian, Yang Yuan. Learning-Based Low-Rank Approximations. NeurIPS 2020
> > >
> > > [2] Chen-Yu Hsu, Piotr Indyk, Dina Katabi and Ali Vakilian. Learning-Based Frequency Estimation Algorithms. ICLR2019
> > >
> > > [3] Simin Liu, Tianrui Liu, Ali Vakilian, Yulin Wan, David P. Woodruff. Learning the Positions in CountSketch. arXiv:2007.09890

---

### Decision · Program_Chairs · 2022-01-20

**Decision:**

Reject

**Comment:**

This paper proposes a new contribution in the recent literature on learning distributions of sketches. While all reviewers have recognized the overall good quality of the presentation, two factors seem to weight heavily on a negative decision: clarifications on the contribution's scope (presented as a tool for general Hessians in the introduction, but ultimately only applied to least-square errors of linear predictors, to recover an explicit factorization of the Hessian matrix) and links with existing literature; weakness of experiments whose small scale does not justify using sketches in the first place. Since this is a "learning" approach, I am particularly sensitive to the latter point, and therefore am inclined to reject, but I encourage the authors to address these two issues with the current draft.